# 3D OBJECT REPRESENTATION LEARNING FOR ROBUST CLASSIFICATION AND POSE ESTIMATION

## ABSTRACT

In this work, we present a novel approach to 3D object representation learning, achieving exceptionally robust classification and 3D pose estimation. Our method uses a 3D representation of object categories utilizing a template mesh, originally introduced for 3D pose estimation, where each mesh vertex is associated with a feature vector. Notably, we establish dense correspondences between image pixels and the 3D template geometry of target object categories by predicting, for each pixel in an image, the feature vector of the corresponding vertex on the template mesh. To achieve viewpoint invariance, we train the feature vectors on mesh vertices by leveraging associated camera poses. During inference, we efficiently estimate the object class by independently matching the vertex features of each template to features from an input image. Intriguingly, we discovered that classification can be accomplished using vertex features alone, without necessitating the use of 3D mesh geometry, thereby speeding up the class label inference process. Our experimental results show better performances of our discriminative 3D object representation compared to conventional 2D representations. It exhibits exceptional robustness across a spectrum of real-world and synthetic out-of-distribution shifts while maintaining competitive accuracy and speed with state-of-the-art architectures on in-distribution data. Importantly, our method stands as the first in the literature to employ a 3D representation for image classification. A major benefit of this explicit representation is the interpretability of its feature space as well as through its individual vertex matching. Additionally, as an extension of a 3D representation initially designed for 3D-pose estimation, our approach is able to perform robust classification and pose estimation jointly and consistently.

## 1 INTRODUCTION

Current computer vision algorithms demonstrate advanced abilities in most visual recognition tasks, such as object recognition, *e.g.,* object classification, detection, and pose estimation (LeCun et al., 1995; Simonyan & Zisserman, 2014; He et al., 2016; Dosovitskiy et al., 2020; Liu et al., 2021). Currently, the most popular approach is to train a deep neural network end-to-end to directly predict the desired output via gradient-based optimization on a specific training set. While this approach yields commendable performance when tested on data that is sampled from a similar distribution as the training data, generalization to out-of-distribution (OOD) scenarios remains a fundamental challenge (Hendrycks & Dietterich, 2019; Michaelis et al., 2019; Zhao et al., 2022; Kortylewski et al., 2020a). In contrast, human vision achieves a significantly better robustness under OOD scenarios, e.g., domain shift, and occlusions (Bengio et al., 2021; Kortylewski et al., 2020b). Cognitive studies hypothesize that human vision relies on a 3D representation of objects while perceiving the world through an analysis-by-synthesis process (Neisser et al., 1967; Yuille & Kersten, 2006). The inherent ability to perceive objects in 3D might enable humans to achieve strong robustness and generalization abilities. Can we achieve a strong generalization in machines by enabling deep neural networks to learn 3D object representations for visual recognition?

In this work, we embed the 3D object geometry explicitly into the neural network architecture, and hence enable the model to learn 3D object representations for object classification. We take inspiration from prior works on 3D pose estimation (Iwase et al., 2021; Wang et al., 2021a), and represent an object category using a 3D template mesh composed of feature vectors at each mesh vertex (Figure 1). For each pixel in a 2D image, our model predicts a feature vector of the corresponding vertex

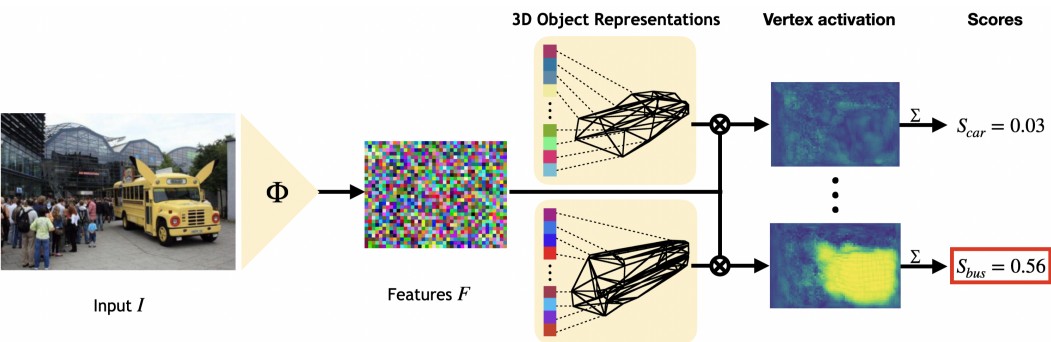

Figure 1: Schematic illustration of the image classification process with our proposed architecture. An image is first processed by a deep network backbone into a feature map $F$. Every object category is represented by a 3D template mesh with feature vectors at each vertex. During classification, the vertex features of each mesh are matched to the feature map to obtain a class prediction score.

in each category template mesh. In this way, the model establishes dense correspondences between image pixels and the 3D template geometry of all target object categories. The feature vectors on the mesh vertices are trained using contrastive learning and associated camera poses to be distinct between different object categories, while also being distinct from the background clutter and being invariant to viewpoint instance-specific details within an object category. During inference, we classify the image by matching the vertex features of each template to the input feature map. In this way, we perform image classification using the vertex features only and without requiring the 3D mesh geometry, hence making the classification inference very efficient. As a byproduct of the inherent 3D representation in our model, we can also estimate the 3D object pose in a two-step manner by first estimating an initial pose via template matching and subsequently refining the estimate using a render-and-compare process to obtain an accurate pose estimate.

We evaluate our approach under real-world OOD shifts on the OOD-CV dataset (Zhao et al., 2023), and synthetic OOD shifts on the corrupted-PASCAL3D+ (Hendrycks & Dietterich, 2019) and occluded-PASCAL3D+(Wang et al., 2021a). Our experiments show that our model is exceptionally more robust compared to other state-of-the-art architectures (both CNNs and Transformers) at object classification while performing on par with in-distribution data in terms of accuracy and inference speed. Despite modifying the original 3D representation, we demonstrate that our 3D pose predictions remain competitive, maintaining performance levels comparable to baseline models explicitly designed for robust 3D pose estimation. In conclusion, our model offers enhanced interpretability through visualizing individual vertex matches, and introduces a novel classification approach with 3D object knowledge, ensuring robustness, and allowing real-time inference—addressing a critical need in the current state-of-the-art landscape.

## 2 RELATED WORK

**Robust Image Classification.** Image Classification is a significant task in computer vision. Multiple influential architectures including ResNet (He et al., 2016), Transformer (Vaswani et al., 2017), and recent Swin-Transformer (Liu et al., 2021) have been designed for this task. However, we observe that these models are not robust in partial occlusion images or out-of-distribution data. Efforts that have been made to close the gap can be mainly categorized into two types: data augmentation and architectural design. Data augmentation includes using a learned augmentation policy (Cubuk et al., 2018), and data mixture (Hendrycks et al., 2019). Architectural changes propose robust pipelines. For instance, (Kortylewski et al., 2020a) proposes an analysis-by-synthesis approach for a generative model to handle occlusions. In addition, challenging benchmarks that use common synthetic corruptions (Hendrycks & Dietterich, 2019) and real out-of-distribution images (Zhao et al., 2022) showed that the performance of standard models drops by a large margin in such scenarios. In this paper, we show that our model is exceptionally robust under occlusions, different corruptions, and real-world OOD images.

**Contrastive learning.** Contrastive learning (Hadsell et al., 2006) is essentially a framework that learns similar/dissimilar representations by optimizing through the similarities of pairs in the repre-

sentation space. Later the idea of using contrastive pair is extended to Triplet (Schroff et al., 2015). While traditional contrastive learning has focused on image-level or text-level representations, the extension to feature-level representations has gained lots of attention after InfoNCE (van den Oord et al., 2019). These works in various fields include SimCLR (Chen et al., 2020), CLIP (Radford et al., 2021), Bai et al. (2022) and so on. In our paper, we adopt a two-level feature contrastive loss that encourages both spatially distinct features and categorical specific features.

**3D representations.** Learning a 3D representation, through matching image features to a 3D model has already been studied (Stark et al., 2010) but were using HOG features of the image following a Proposal-Validation process which is slow. In contrast, Choy et al. (2015); Zeeshan Zia et al. (2013) introduced a detection method for object parts, offering a potential avenue for 3D pose estimation based on spatial layouts. Our approach differs from using HOG features, opting for neural features capable of encoding richer information. Furthermore, we aim to eliminate the need for object-parts annotations in the learning process for our representation. In that direction, render-and-compare methods optimize the predicted pose by reducing the reconstruction error between 3D-objects projected representations and the extracted representations. It can be seen as an approximate analysis-by-synthesis (Grenander, 1970) approach, which has is much more robust against OOD data at 3D pose estimation (Wang et al., 2021a;b; Iwase et al., 2021) compared to classical discriminative methods (Tulsiani & Malik, 2015; Mousavian et al., 2017; Zhou et al., 2018). In particular, NeMo (Wang et al., 2021a) introduced a neural 3D representation trained with contrastive loss within each object category for pose estimation. Our work builds on and substantially extends NeMo in multiple ways: (1) An architecture with a class-agnostic backbone, while NeMo uses a separate backbone for each class. This enables us to introduce a class-contrastive loss for object classification, while NeMo is not capable of classifying images. (2) A principled, efficient way of performing classification inference, which does not rely on the demanding render-and-compare technique by considering vertices collectively as a mesh, but rather exploit individual vertices. (3) A comprehensive mathematical formulation that derives a vMF-based contrastive training loss. Combining these three points, we achieve substantial OOD robustness in classification, while performing on par with models that were specifically designed for robust pose estimation, such as NeMo. Additionally, we note the presence of one **concurrent work**—NTDM (Wang et al., 2023)—on arXiv which also extends NeMo to allow classification. However, NDTM focuses on deformable geometry of object, and only marginally explore classification. Our method outperforms it significantly in all scenarios (see Apprendix B.5) while performing classification robustly and efficiently, whereas NTDM performs inference at a 25 slower rate by adopting a *brute force search* using render-and-compare for each object category during inference, and hence making it practically infeasible.

## 3   3D OBJECT REPRESENTATION LEARNING

In this section, we present our framework for 3D object representation learning. In the following, we describe our 3D object representation, how we learn its parameters, and how we perform inference.

### 3.1   3D OBJECT REPRESENTATION

We use an explicit representation of the 3D object geometry. This includes a cuboid mesh for each object category that encompasses the variable geometries of individual object instances in the category (Wang et al., 2021a). Furthermore, each mesh vertex has an associated feature vector. More formally, the 3D object representation for an object category $y$ is given by $\mathfrak{N}_y = \{\mathcal{V}_y, \mathcal{C}_y\}$, where $\mathcal{V}_y = \{V_k \in \mathbb{R}^3\}_{k=1}^{R_y}$ is the set of vertices and $\mathcal{C}_y = \{C_k \in \mathbb{R}^D\}_{k=1}^{R_y}$ is the set of vertex features. The remainder of the image (i.e. outside the object) is represented by a set of background features $\mathcal{B} = \{\beta_n \in \mathbb{R}^D\}_{n=1}^{N}$ where $N$ is a pre-fixed hyperparameter. This set of background features $\mathcal{B}$ are shared for all object categories. We define the set of vertex features from all object categories as $\mathcal{C} = \{\mathcal{C}_y\}_{y=1}^{Y}$, where $Y$ is the total number of object categories .

Our model uses a feature extractor $\Phi_w(I) = F \in \mathbb{R}^{D \times H \times W}$ to obtain image features from input image $I$, where $w$ denotes the parameters of the CNN backbone. The backbone output is a feature map $F$ with feature vectors $f_i \in \mathbb{R}^D$ at positions $i$ on a 2D lattice.

To obtain the correspondence between a vertex $r$ of an object mesh and a location $i$ in the feature map, we adopt a perspective projection. Given a rotation matrix $\mathbf{R}$, the extrinsic camera parameters

translation $\mathbf{T}$ and camera pose $\alpha$, the 2D projection of a vertex on the mesh onto the image is computed as:

$$v_r = (\mathbf{R} \cdot V_r + \mathbf{T}) \cdot \mathbf{P}, \tag{1}$$

where $\mathbf{P}$ contains the camera intrinsic parameters. We assume that the parameters for the perspective projection are known when learning but will need to be inferred by our model at test time. Throughout the remaining paper, we denote $f_{r \to i}$ to indicate the extracted feature $f_i$ at location $i$ that vertex $V_r$ projects to.

We relate the extracted features to the vertex and background features by von-Mises-Fisher (vMF) probability distributions. In particular, we model the probability of generating the feature $f_{r \to i}$ from corresponding vertex feature $C_r$ as $P(f_{r \to i}|C_r) = c_p(\kappa)e^{\kappa f_{r \to i} \cdot C_r}$, where $C_r$ is the mean of each vMF kernel, $\kappa$ is the corresponding concentration parameter, and $c_p$ is the normalization constant ($\|f_{r \to i}\| = 1, \|C_r\| = 1$). We also model the probability of generating the feature $f_i$ from background features as $P(f_i|\beta_n) = c_p(\kappa)e^{\kappa f_i \cdot \beta_n}$ for $\beta_n \in \mathcal{B}$. When learning the models, as described in the next section, we will learn the vertex features $\mathcal{C}$, the background features $\mathcal{B}$, and the parameters $w$ of the neural network backbone. Our model requires that the backbone is able to extract features that are invariant to the viewpoint of the object to ensure that $f_i \cdot C_r$ is large irrespective of the viewpoint.

## 3.2 3D OBJECT-CONTRASTIVE REPRESENTATION LEARNING

Learning our model is challenging because we not only need to learn the likelihood functions $P(f_{r \to i}|C_r)$ and $P(f_i|\mathcal{B})$, but also the parameters $w$ of the backbone. We need to train the backbone to ensure that the vertex features are viewpoint-invariant, otherwise we could use an off-the-shelf backbone and train the model by maximum likelihood.

In detailed formulation, for any extracted feature $f_{r \to i}$, we maximize the probability that the feature was generated from $P(f_{r \to i}|C_r)$ instead of from any other alternatives. This motivates us to use contrastive learning where we compare the probability that an extracted feature $f_{r \to i}$ is generated by the correct mesh vertex $V_r$ or from one of three alternative processes, namely, (i) from non-neighboring vertex features of the same object (ii) from the vertex features of other objects, and (iii) from the background features:

$$\frac{P(f_{r \to i}|C_r)}{\sum_{\substack{C_l \in \mathcal{C}_y \\ C_l \notin \mathcal{N}_r}} P(f_{r \to i}|C_l) + \omega_\beta \sum_{\beta_n \in \mathcal{B}} P(f_{r \to i}|\beta_n) + \omega_{\bar{y}} \sum_{C_m \in \mathcal{C}_{\bar{y}}} P(f_{r \to i}|C_m)} \tag{2}$$

where $\mathcal{N}_r = \{V_k : \|\mathbf{V}_r - \mathbf{V}_k\| < D, r \neq k\}$ is the neighborhood of $V_r$, $D$ is a threshold controlling the size of neighborhood. $y$ is the category of the image and $\bar{y}$ is a set of all other categories except $y$. $\omega_\beta = \frac{P(\beta_n)}{P(C_r)}$ is the ratio of the probability that an image pixel corresponds to the background instead of to the vertex $V_r$, and $\omega_{\bar{y}} = \frac{P(C_m)}{P(C_r)}$ is the ratio of the probability that an image pixel corresponds to vertex of other categories instead of to the vertex $V_r$. During training, the ground-truth pose specifies the correspondence between object vertices and image pixels.

Then we compute the final loss $\mathcal{L}(\mathcal{C}, \mathcal{B}, w)$ by taking the logarithm and summing over all training examples – the set of features $\{f_{r \to i}\}$ from the training set

$$-\sum_{r \in R_v} o_r \cdot \log\left(\frac{e^{\kappa f_{r \to i} \cdot C_r}}{\sum_{\substack{C_l \in \mathcal{C}_y \\ C_l \notin \mathcal{N}_r}} e^{\kappa f_{r \to i} \cdot C_l} + \omega_\beta \sum_{\beta_n \in \mathcal{B}} e^{\kappa f_{r \to i} \cdot \beta_n} + \omega_{\bar{y}} \sum_{C_m \in \mathcal{C}_{\bar{y}}} e^{\kappa f_{r \to i} \cdot C_m}}\right) \tag{3}$$

where $o_r$ takes value 1 if the vertex $V_r$ is visible and 0 otherwise. It is computed from the object geometry and the annotated pose. In our experiments, we fix the concentration parameter $\kappa$ to be constant and do not learn them.

**Updating Vertex and Background Features.** The features, $\mathcal{C}$ and $\mathcal{B}$ are updated after every gradient-update of the feature extractor. Following He et al. (2020), we use momentum update for the vertex features:

$$C_r = C_r * \gamma + f_{r \to i} * (1 - \gamma), \|C_r\| = 1, \tag{4}$$

The background features, are simply resampled from the newest batch of training images. In particular, we remove the oldest features in $\mathcal{B}$, i.e., $\mathcal{B} = \{\beta_n\}_{n=1}^N \setminus \{\beta_n\}_{n=1}^T$.
Next, we randomly sample $T$ new background features $f_b$ from the feature map, where $b$ is location that no vertex projected to, and add them into the background feature set $\mathcal{B}$, *i.e.,* $\mathcal{B} \leftarrow \mathcal{B} \cup \{f_b\}$.
We note that $\gamma$ and $T$ are hyper-parameters of our model.

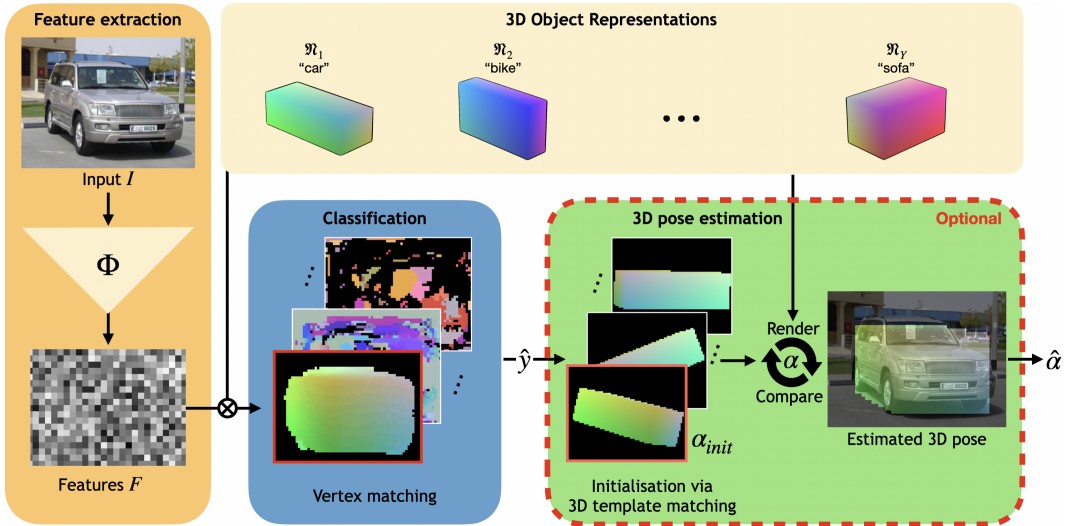

Figure 2: An overview of the full *inference pipeline*. Our proposed architecture is composed of a backbone $\Phi$ and a set of 3D object representations $\{N_y\}$, where each category is represented as a cuboid mesh and associated features at each mesh vertex. During inference, an image is first processed by the backbone into a feature map $F$. Subsequently, the object class is predicted by matching individually the vertex features to the feature map **without** taking into account the cuboid geometry (blue box). We color-code the detected vertices for each object class to highlight the interpretability of our method. Optionally, for the predicted object class $\hat{y}$ we can further estimate the object pose in a two-step process, by first obtaining an initial estimate via template matching and subsequently refinement using render-and-compare.

## 3.3 INFERENCE OF OBJECT CATEGORY AND 3D POSE

Our complete inference pipeline is illustrated in Figure 2 and discussed in detail in the following.

**Object Classification via Feature Matching without Geometry.** We perform classification in a fast and robust manner via matching the extracted features to the learned vertex features and background features. In short, for each object category $y$, we compute the foreground likelihood $P(f_i|\mathcal{C}_y)$ and the background likelihood $P(f_i|\mathcal{B})$ on all location $i$. In this process, we do not take into account the mesh geometry, and reduces the matching to a simple convolution operation, hence making it very fast. To classify an image, we compare the total likelihood scores of each class average through all location $i$.

In detail, we define a new binary valued parameter $z_{i,r}$ such that $z_{i,r} = 1$ if the feature vector $f_i$ matches best to any vertex feature $\{C_r\} \in \mathcal{C}_y$, and $z_{i,r} = 0$ if it matches best to a background feature. The object likelihood of the extracted feature map $F = \Phi_w(I)$ can then be cumputed as:

$$\prod_{f_i \in F} P(f_i|z_{i,r}, y) = \prod_{f_i \in F} P(f_i|C_r)^{z_{i,r}} \prod_{f_i \in F} \max_{\beta_n \in \mathcal{B}} P(f_i|\beta_n)^{1-z_{i,r}} \tag{5}$$

As described in Section 3.1, the extracted features follow a vMF distribution. Thus the final prediction score of each object category $y$ is:

$$S_y = \sum_{f_i \in F} max\{ \max_{C_r \in \mathcal{C}_y} f_i \cdot C_r, \max_{\beta_n \in \mathcal{B}} f_i \cdot \beta_n \} \tag{6}$$

The final category prediction is $\hat{y} = argmax\{S_y, y \in Y\}$. Figure 2 (blue box) illustrates the vertex matching process for different object classes by color coding the detected vertices. We can observe that for the correct class, the vertices can be detected coherently even without taking the mesh geometry into account (as can be observed by the smooth color variation), while for wrong classes this is not the case. Our ability to visualize the predicted vertex correspondence also demonstrates an advanced interpretability of the decision process compared to standard deep network classifiers.

**Render-and-Compare for Pose Estimation and Fine-tuning.** Given the predicted object category $\hat{y}$, we use the mesh $\mathfrak{N}_{\hat{y}}$ to estimate the camera pose $\alpha$. This method is beneficial particularly since it can exploit the 3D structure-related information stored in the vertex features.

We maximize $P(f_i|C_r)$ by optimizing over pose $\alpha$ via render-and-compare. At each optimization step, we use the projection under pose $\alpha$ to determine whether each pixel $i$ belongs to the foreground or background. Specifically, $F_{front} = \{f_{r \to i} \in F, o_r = 1 \, \forall \, r \in \mathcal{V}_{\hat{y}}\}$ is the set of all projected and visible extracted features, and $F_{back} = F \setminus F_{front}$:

$$\prod_{f_i \in F} P(f_i|v, \hat{y}) = \prod_{\substack{C_r \in \mathcal{C}_{\hat{y}} \\ f_{r \to a} \in F_{front}}} P(f_{r \to a}|C_r) \prod_{f_b \in F_{back}} \max_{\beta_n \in \mathcal{B}} P(f_b|\beta_n) \tag{7}$$

Following the vMF distribution, we optimize the pose $\alpha$ with a feature reconstruction loss:

$$\mathcal{L}(\alpha) = \sum_{\substack{C_r \in \mathcal{C}_{\hat{y}} \\ f_{r \to a} \in F_{front}}} f_{r \to a} \cdot C_r + \sum_{f_b \in F_{back}} \max_{\beta_n \in \mathcal{B}} f_b \cdot \beta_n \tag{8}$$

We estimate the pose as shown in the green section of the Figure 2. We first find the best initialization of the object pose $\alpha$ by computing the reconstruction loss (Eq.8) for a set of pre-defined poses via template matching. Subsequently, we start gradient-based optimization using the initial pose that achieved the lowest loss to obtain the final pose prediction $\hat{\alpha}$.

## 4 EXPERIMENTS

In this section, we discuss our experimental setup (Section 4.1), present baselines and results for classification (Section 4.2) and 3D pose estimation (Section 4.3). Additionally, we perform in-depth evaluations of interpretability and prediction consistency, and an ablation study (Section 4.4).

### 4.1 SETUP

**Datasets**. We can thoroughly test our model's robustness under OOD scenarios using four datasets: PASCAL3D+ (P3D+)(Xiang et al., 2014), occluded-PASCAL3D+(Wang et al., 2020), corrupted-PASCAL3D+(Hendrycks & Dietterich, 2019; Michaelis et al., 2019), and Out-of-Distribution-CV-v2 (OOD-CV)(Zhao et al., 2023). PASCAL3D+ includes 12 object categories with 3D annotations. We split the dataset into a training set of 11045 images and a testing set with 10812 images, referred to as L0. Building on the PASCAL3D+ dataset, the occluded-PASCAL3D+ dataset is a test benchmark that evaluates robustness under multiple levels of occlusion. It simulates realistic occlusion by superimposing occluders on top of the objects with three different levels: L1: 20%-40%, L2: 40%-60%, and L3:60%-80%, where each level has corresponding percent of objects occluded. Corrupted-PASCAL3D+ corresponds to PASCAL3D+ on which we apply 12 types of corruptions (Hendrycks & Dietterich, 2019; Michaelis et al., 2019) to each image of the original P3D+ test dataset. We choose a severity level of 4 out of 5 for each applied corruption. The OOD-CV-v2 dataset is a benchmark that includes real-world OOD examples of 10 object categories varying in terms of 6 nuisance factors: pose, shape, context, occlusion, texture, and weather.

**Implementation Details.** Each 3D template mesh contains approximately 1100 vertices that are distributed uniformly on the cuboid. The shared feature extractor $\Phi$ is a ResNet50 model with two upsampling layers and an input shape of $640 * 800$. All features have a dimension of $D = 128$ and the size of the feature map $F$ is $\frac{1}{8}$ of the input size. Our 3D object representation has been learned collectively using a contrastive approach as described in 3.2, taking around 20 hours on 4 RTX 3090 under 200 epochs. During training, we keep track of all vertex features and $N = 2560$ background features. For each gradient step, we use $\gamma = 0.5$ for momentum update of the vertex features and sample $T = 5$ new background features from the background of the image to update $\mathcal{B}$. We set $\kappa = 1$ (see Appendix C for more details). During inference (Section 3.3), we firstly predict the object class that corresponds to the highest affinity. The feature matching for classification takes around **0.01**s per sample on 1 RTX 3090, which is comparable to cutting-edge classification

models. For 3D-pose estimation, we leverage the full geometry knowledge. Hence, we apply render-and-compare to render the mesh model and compare it to extracted feature map $F$. For initializing the pose estimation, we follow Wang et al. (2021a) and sample $144$ poses ($12$ azimuth angles, $4$ elevation angles, $3$ in-plane rotations) and choose the pose with lowest feature reconstruction loss as initialization. We then minimize the loss using gradient-descent (Equation 8) to get a better estimate of the object pose. Pose inference pipeline takes around $0.21$s per sample on 1 RTX 3090.

**Evaluation.** We evaluate our approach on two tasks separately: classification and pose estimation. The 3D pose estimation involves predicting azimuth, elevation, and in-plane rotations of an object with respect to a camera. Following (Zhou et al., 2018), the pose estimation error is calculated between the predicted rotation matrix $R_{\text{pred}}$ and the ground truth rotation matrix $R_{\text{gt}}$ as $\Delta\left(R_{\text{pred}}, R_{\text{gt}}\right) = \left\|\log m\left(R_{\text{pred}}^T R_{\text{gt}}\right)\right\|_F / \sqrt{2}$. We measure accuracy using two thresholds $\frac{\pi}{18}$ and $\frac{\pi}{6}$.

**Baselines.** We describe in the following the baselines we use to compare with our approach, which have proven their robustness in OOD scenarios (Liu et al., 2022; Wang et al., 2021a).
*Classification.* We compare the performance of our approach to four other baselines (*i.e.,* Resnet50, Swin-T, Convnext, and ViT-b-16) for the classification task. For each baseline, we replaced the original head with a classification head for which the output is the number of classes in the dataset (*i.e.,* 12 for (occluded,corrupted)-P3D+; 10 for OOD-CV). We finetune each baselines during 100 epochs. In order to make baselines more robust, we apply some data augmentation (i.e., scale, translation, rotation, and flipping) for each baseline during training.
*3D-Pose estimation.* We compare the performance of our approach to five other baselines for the 3D pose estimation task. For Resnet50, Swin-T, Convnext, and ViT-b-16, we consider the pose estimation problem as a classification problem by using 42 intervals of $\sim 8.6°$ for each parameter that needs to be estimated (azimuth and elevation angle, and in-plane rotation). We finetune each baseline for 100 epochs. Similarly to classification, we apply data augmentation to make the baselines more robust while we do not employ any form of image augmentation in our approach. For the remaining baseline, which was designed for robust 3D pose estimation, we used the methodology described in Wang et al. (2021a) for both training and evaluation of NeMo using the publicly available code (similar score to the ones reported as *NeMo-SingleCuboid*) by training a NeMo model for each class.

## 4.2 ROBUST OBJECT CLASSIFICATION

We first evaluate the performance of our model on IID data. As the L0 (clean images) column of Table 1 shows, our approach achieves almost a perfect score (*i.e.,* $99.5\%$) for classification on IID data, which is comparable to other baslines. Furthermore, our approach manages to robustly classify images in various out-of-distribution scenarios. From Table 1, we can see that our representation allows to outperform all other traditional baselines with around 6% accuracy on average for different levels of occlusions and with up to 33% accuracy boost for images under six different types of nuisances in OOD-CV. For corrupted data, our approach performs better on average while performing worse for some corruptions (see Appendix S3). We can easily explain that by the fact that some baselines have been pretrained on massive amounts of data (e.g., ConvNext, ViT), sometimes including these corruptions. As a result, the comparison might not be fair. Despite this consideration, our appraoch performs better on average compared to all baselines. Based on these evidences, our approach has made a **great improvement in OOD** scenarios while **maintaining cutting-edge accuracy for IID** data for classification. Finally, it is also worth noting that our approach is **much more consistent** than all baselines (i.e., results' standard deviations are 15.8 and 9.7 for ViT and Ours, respectively). Independently of the nuisance's nature, our approach tends to have consistent performances, which indicates the intrinsic robustness in our architecture.

## 4.3 ROBUST 3D POSE ESTIMATION

In order to estimate the 3D pose, our approach uses $\alpha_{init}$, which consists of a coarse prediction by computing the similarity (Equation 8) with each of the 144 pre-rendered maps, as initialization for the render-and-compare procedure. The final prediction $\hat{\alpha}$ is therefore the pose that maximizes Equation 7 (we also perform an evaluation of the quality of our initialization in Appendix S6) during a render-and-compare procedure. According to the results in Table 2, our approach outperforms all feed-forward baselines significantly across all datasets. In addition, our approach competes with

Table 1: Classification accuracy results on P3D+, occluded-P3D+, OOD-CV-v2 and corrupted-P3D+ datasets. First is highlighted in **bold**, second is underlined. L0 corresponds to unoccluded images from Pascal3D+, and occlusion levels L1-L3 are from occluded-P3D+ dataset with occlusion ratios stated in 4.1. Our approach performs similarly in IID scenarios, while steadily outperforming all baselines in OOD scenarios. Full results can be found in Appendix.

| Dataset | P3D+ | occluded-P3D+ | | | | OOD-CV | corrupted-P3D |
|---|---|---|---|---|---|---|---|
| Nuisance | L0 | L1 | L2 | L3 | Mean | Mean | Mean |
| Resnet50 | 99.3 | 93.8 | 77.8 | 45.2 | 72.3 | 51.4 | 78.7 |
| Swin-T | 99.4 | 93.6 | 77.5 | 46.2 | 72.4 | 64.2 | 78.9 |
| Convnext | 99.4 | 95.3 | 81.3 | 50.9 | 75.8 | 56.0 | 85.6 |
| ViT-b-16 | 99.3 | 94.7 | 80.3 | 49.4 | 74.8 | 59.0 | 87.6 |
| Ours | **99.5** | **97.2** | **88.3** | **59.2** | **81.6** | **85.2** | **91.3** |

Table 2: 3D-Pose Estimation results for different datasets. A prediction is considered correct if the angular error is lower than a given threshold (*i.e.,* $\frac{\pi}{6}$, and $\frac{\pi}{18}$). Higher is better. Our approach shows it is capable of robust 3D pose estimation that performs similarly to the current state-of-the-art.

| Dataset | P3D+ | occluded-P3D+ | corrupted-P3D+ | OOD-CV | P3D+ | occluded-P3D+ | corrupted-P3D+ | OOD-CV |
|---|---|---|---|---|---|---|---|---|
| Threshold | | | $\pi/6$ | | | | $\pi/18$ | |
| Resnet50 | 82.2 | 53.8 | 33.9 | 51.8 | 39.0 | 15.8 | 15.8 | 18.0 |
| Swin-T | 81.4 | 48.2 | 34.5 | 50.9 | 46.2 | 16.6 | 15.6 | 19.8 |
| Convnext | 82.4 | 49.3 | 37.1 | 50.7 | 38.9 | 14.1 | 24.1 | 19.9 |
| ViT-b-16 | 82.0 | 50.8 | 38.0 | 48.0 | 38.0 | 15.0 | 21.3 | 21.5 |
| NeMo | 86.1 | **62.2** | **48.0** | 71.4 | **61.0** | **31.8** | 43.4 | **21.9** |
| Ours | **86.2** | 60.9 | 47.9 | **71.5** | 60.5 | 31.6 | 43.4 | 21.8 |

NeMo (Wang et al., 2021a), the current state-of-the-art method for robust 3D pose estimation, despite not being explicitly designed for robust 3D pose estimation.

## 4.4 COMPREHENSIVE ASSESSMENT OF OUR REPRESENTATION

**Interpretability.** Our explicit 3D representation can be leveraged to retrieve the object class and its pose in an image. Very insightful information also lies in the vertex matching between image features and the 3D mesh representation. By visualizing matches (see Figure 3c-d and more examples including videos in Appendix D), we understand better which part of the object is visible. As illustrated in Figure 3d, the bottom part of the bottle, which is occluded by the dog, doesn't have color, meaning that no image features were matched with these vertices. Hence, our 3D representation provides a straightforward way of visualizing which part of the object is visible and therefore, which part of the object is occluded. T-SNE plots in Figure 3a-b show that our features are disentangled and encode useful information in terms of object classes (different cluster for each category in Fig. 3a) or 3D pose (consistent distribution of car instances depending of their pose in Fig. 3b).

**Consistency.** Some exciting characteristics of our approach lie in the fact that it is consistent between the different tasks. If one wants a model able to solve multiple tasks, we expect it to be consistent between the predictions for all tasks, as a human would do (*i.e.,* if the scene is really tricky we expect all predictions to be of worse quality in a consistent manner). When we have a closer a look at the Table 4, we observe that results for 3D Pose estimation, and the full pipeline (the full pipeline outputs in a first step the class $\hat{y}$ and then use this prediction to estimate the pose $\hat{\alpha}$ for this object category) are fairly similar. In IID scenarios, we difference is only of $0.2\%$, while in OOD scenarios the difference is around $1\%$ on average. We believe, this consistency comes from the common explicit 3D representation that is being shared for all tasks. Such a behavior would not be expected between different task-specific models that are trained separately.

**Efficiency.** For classification, our method matches real-time performance as other CNN or transformer-based baselines, handling over 50FPS. Despite variations in parameter numbers among baselines (Swin-T: 28M, ViT-b-16: 86M, Ours: 83M), we find no correlation of the parameter count with OOD robustness. For pose estimation, compared to render-and-compare-based methods, our

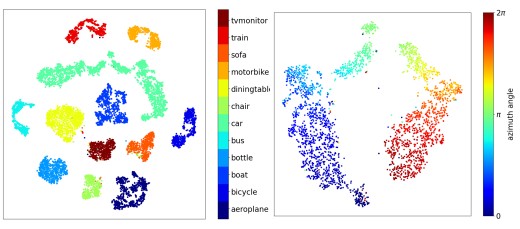

(a) t-SNE of classes    (b) t-SNE of azimuth poses

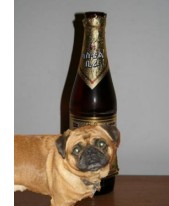 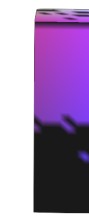

(c) Input image    (d) Mesh Vertex Activation

Figure 3: (a-b) t-SNE plots showing interpretability of our features. (c-d) Qualitative results showing that visible vertices are matched while occluded vertices are not matched.

Table 4: Consistency of predictions over classification and 3D pose estimation. Specifically, column 3 show the performance on the accuracy of those images with both object pose and class correctly predicted, w.r.t. accuracy on each task respectively.

| Dataset | P3D+ | occluded-P3D+ | | | |
|---|---|---|---|---|---|
| Task | L0 | L1 | L2 | L3 | Mean |
| Classification | 99.5 | 97.2 | 88.3 | 59.2 | 81.6 |
| 3D Pose (threshold $\frac{\pi}{6}$) | 86.2 | 77.6 | 66.1 | 41.9 | 61.9 |
| Classification & 3D Pose | 86.0 | 77.0 | 65.4 | 39.9 | 60.8 |

Table 5: Ablation studies on PASCAL3D+ and Occluded PASCAL3D+.

| Components | | P3D+ | occluded-P3D+ | | | |
|---|---|---|---|---|---|---|
| $\mathcal{B}$ | Object shape | L0 | L1 | L2 | L3 | Mean |
| x | single feature | 93.2 | 90.3 | 80.4 | 54.0 | 74.9 |
| ✓ | sphere | 99.3 | 97.0 | 87.9 | 59.0 | 81.3 |
| x | cuboid | 99.3 | 97.0 | 85.7 | 53.0 | 78.6 |
| ✓ | cuboid | **99.5** | **97.2** | **88.3** | **59.2** | **81.6** |

model uses a significantly lower parameter count (Ours: 83M, NeMo: 996M) due to our class-agnostic backbone. We also observe that our model converges faster in the render-and-compare optimization compared to NeMo (Ours: 30 steps, NeMo: 300 steps), which can be attributed to our class-contrastive representation. More detailed comparisons can be found in Appendix B.6.

**Ablations.** To evaluate the significance of our approach's components, we conduct the following ablations. In our approach, we represent our objects using a cuboid mesh. In alternative approaches, we could choose to (1) employ a finer-grained mesh, (2) utilize a single feature vector to represent the entire image content, or (3) adopt a generic spherical mesh that is the same for all object classes. Opting for the first alternative would necessitate either establishing a deformable mapping between fine-grained meshes for each sub-category, which is beyond the scope of this work. In Table 5, we explore ablations related to object shape of our 3D representation. As expected, the choice of mesh shape does not exert a pronounced influence on performance. Whether employing a cuboid or a spherical mesh, performance remains relatively similar. However, we do observe a slight advantage in favor of the cuboid shape (more details in Appendix B.7.1). This preference may be attributed to the cuboid's closer approximation of the objects' true shapes. Additionally, we considered a "single vertex" approach, where a single feature vector per class is employed during training using contrastive learning (more details in Appendix B.7.2). We observe a performance drop by up to 9%, which highlights the importance of the object geometry. These findings corroborate the classification results of our method: by selectively omitting some geometric information (*i.e.*, the 3D structure), we can reach similar outcomes while significantly enhancing computational efficiency and reducing memory requirements. Ultimately, we note that the background model $\mathcal{B}$ is beneficial during training since it promotes greater dispersion among neural features. This proves to be useful for inference in cases marked by occlusions but does not have visible effect in IID scenarios.

## 5 CONCLUSION

In this work, we demonstrate the benefit of 3D object representations for robust classification and pose estimation. We explicitly introduce a 3D object geometry, *i.e.,* a mesh with features on vertices, into the architecture of deep neural networks for image classification and present a contrastive learning framework to learn this representation from class labels and annotated 3D object poses. Our experimental results show that 3D object representations achieve higher robustness compared to other state-of-the-art architectures under out-of-distribution scenarios, *e.g.,* occlusion and corruption, with competitive performance in in-distribution scenarios, and similar inference speed for image classification. Further, experiments demonstrate that our approach can also estimate 3D pose accurately, achieves an enhanced interpretability and consistency of multi-task predictions.

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

## SUPPLEMENTARY MATERIAL FOR PAPER "3D OBJECT REPRESENTATION LEARNING FOR ROBUST CLASSIFICATION AND POSE ESTIMATION"

We provide additional results and discussions to support the experimental results in the main paper. We also study in more details the influence and sensitivity of our experimentally fixed thresholds on the results.

## A NeMo BASELINE EXTENSION

In the following, we introduce in more details how Nemo (Wang et al., 2021a) can be naïvely extended to perform classification and how does it perform for this new task.

### A.1 EXTENSION PROCEDURE

NeMo (Wang et al., 2021a) is originally designed to perform 3D-pose estimation and the class of the object is considered as known. We leverage one class-specific trained NeMo model for each object category (12 for Pascal3D+; 10 for OOD-CV). We use the exact same procedure as in the original paper(Wang et al., 2021a) for training the class-specific NeMo models and for inference. Since NeMo relies on a render-and-compare approach, we can obtain the reconstruction loss from the final prediction for each candidate class which we leverage to assess the quality of the predicted 3D-pose. Finally, the class corresponding to the lowest loss corresponds to the predicted class.

### A.2 CLASSIFICATION

In this section, we compare results between the extended-NeMo (as described in the previous section) and our approach for classification. Table S1 shows that for classification, our approach considerably outperforms the naïve extension of NeMo for Pascal3D+ and OOD-CV. In the most challenging scenarios (e.g., L3 occlusion, weather), our approach even performs more than 2x better. Besides this substantial performance improvement, the computation requirements are notably lower (both in terms of memory and temporal requirements). Hence, this demonstrates that learning the neural textures in a discriminative manner between classes is crucial to observe good classification performances.

Table S1: Classification accuracy results on PASCAL3D+, occluded-PASCAL3D+ and OOD-CV datasets. Higher is better. We observe a significant performance improvement between the naïve NeMo extension (referred to as "ext.-Nemo") and our approach. Our approach performs better in all scenarios.

| Dataset | P3D+ | occluded-P3D+ | | | | OOD-CV | | | | | | |
|---|---|---|---|---|---|---|---|---|---|---|---|---|
| Nuisance | L0 | L1 | L2 | L3 | Mean | Context | Pose | Shape | Occlusion | Texture | Weather | Mean |
| ext.-NeMo | 88.0 | 72.5 | 49.3 | 22.3 | 48.0 | 52.2 | 43.2 | 54.8 | 47.6 | 45.5 | 40.4 | 46.3 |
| Ours | **99.5** | **97.2** | **88.3** | **59.2** | **81.6** | **85.3** | **88.1** | **83.6** | **81.2** | **90.1** | **82.8** | **85.2** |

### A.3 3D POSE ESTIMATION

3D Pose estimation is not affected by any changes of the extension procedure. Hence, results reported in the main paper for NeMo still stand for the extended version of NeMo.

## B ADDITIONAL RESULTS

In this section, we supplement the findings outlined in the main paper and show results for $\frac{\pi}{18}$ and $\frac{\pi}{6}$ thresholds. Then we show the full results for all tasks for all corruptions of the corrupted-PASCAL3D+ dataset.

## B.1 CLASSIFICATION

Tables S3 and S3 show complementary results to the ones shown in the paper. We observe similar trends for all metrics than the ones presented in Sections 1. Our approach shows much higher performances compared to current state-of-the-art (SOTA) for the classification task.

Table S2: Classification accuracy results on PASCAL3D+, occluded-PASCAL3D+ and OOD-CV-v2 datasets. First is highlighted in **bold**, second is underlined. L0 corresponds to unoccluded images from Pascal3D+, and occlusion levels L1-L3 are from occluded-PASCAL3D+ dataset with occlusion ratios stated in 4.1. Our approach performs similarly in IID scenarios, while steadily outperforming all baselines in OOD scenarios.

| Dataset | P3D+ | occluded-P3D+ | | | | OOD-CV | | | | | | |
|---|---|---|---|---|---|---|---|---|---|---|---|---|
| Nuisance | L0 | L1 | L2 | L3 | Mean | Context | Pose | Shape | Occlusion | Texture | Weather | Mean |
| Resnet50 | 99.3 | 93.8 | 77.8 | 45.2 | 72.3 | 45.1 | 61.2 | 55.2 | 69.5 | 48.3 | 47.3 | 51.4 |
| Swin-T | 99.4 | 93.6 | 77.5 | 46.2 | 72.4 | 63.0 | 71.4 | 65.9 | 69.9 | 61.4 | 59.6 | 64.2 |
| Convnext | 99.4 | 95.3 | 81.3 | 50.9 | 75.8 | 53.6 | 61.2 | 60.8 | 72.9 | 57.2 | 47.1 | 56.0 |
| ViT-b-16 | 99.3 | 94.7 | 80.3 | 49.4 | 74.8 | 57.8 | 67.3 | 61.0 | 72.3 | 54.7 | 54.5 | 59.0 |
| Ours | **99.5** | **97.2** | **88.3** | **59.2** | **81.6** | **85.3** | **88.1** | **83.6** | **81.2** | **90.1** | **82.8** | **85.2** |

Table S3: Classification accuracy results on corrupted-PASCAL3D+ under 12 different types of common corruptions. Our approach outperforms the baseline network that it was built upon (ResNet50) by a wide margin, and also outperforms other state-of-the-art architectures on average.

| Dataset | P3D+ | corrupted-P3D+ | | | | | | | | | | | | |
|---|---|---|---|---|---|---|---|---|---|---|---|---|---|---|
| Nuisance | L0 | defocus blur | glass blur | motion blur | zoom blur | snow | frost | fog | bright-ness | con-trast | elastic trans. | pixelate | jpeg | mean |
| Resnet50 | 99.3 | 67.6 | 41.4 | 73.5 | 87.5 | 84.4 | 84.3 | 93.9 | 98.0 | 90.0 | 46.4 | 82.1 | 95.5 | 78.7 |
| Swin-T | 99.4 | 60.7 | 37.1 | 70.9 | 81.3 | 88.5 | 91.6 | 95.4 | 97.9 | 92.1 | 56.3 | 79.2 | 95.3 | 78.9 |
| Convnext | 99.4 | 70.1 | 58.7 | 76.5 | **90.0** | **92.3** | 92.9 | **98.5** | **99.2** | **98.4** | 67.6 | 84.2 | **98.7** | 85.6 |
| ViT-b-16 | 99.3 | 64.5 | **78.1** | 80.3 | 88.2 | 91.2 | **94.1** | 90.5 | 98.7 | 85.1 | 84.8 | 96.9 | **98.7** | 87.6 |
| Ours | **99.5** | **90.5** | 65.7 | **86.4** | 84.2 | 91.2 | 89.5 | 98.4 | 98.4 | 97.1 | **97.2** | **97.1** | 98.4 | **91.3** |

## B.2 ADDITIONAL RESULTS FOR $\frac{\pi}{18}$ AND $\frac{\pi}{6}$ THRESHOLDS

Table S4 shows complementary results to the ones shown in the paper. We observe similar trends for all metrics than the ones presented in Sections 4.3. Our approach shows equivalent performances to current state-of-the-art (SOTA) for the 3D-pose estimation task even though it is not specifically designed to perform this task.

## B.3 FULL CORRUPTED-PASCAL3D+ RESULTS

Table S5 shows corrupted-PASCAL3D+ results for all studied tasks and for all corruptions. Similarly to what have been discussed previously, our approach significantly outperforms all baselines for Classification. For 3D-pose estimation, we observe competitive performances compared to current SOTA that have been specifically designed for this task.

Table S4: Pose Estimation results on (occluded)-PASCAL3D+, and OOD-CV dataset. Pose accuracy is evaluated for error under two thresholds: $\frac{\pi}{6}$ and $\frac{\pi}{18}$ separately. Noticeably, our approach has equivalent performances to current SOTA for 3D-pose estimation event hough it has not been specifically designed for this task.

| Dataset | | P3D+ | occluded-P3D+ | | | | OOD-CV | | | | | | |
|---|---|---|---|---|---|---|---|---|---|---|---|---|---|
| Nuisance | | L0 | L1 | L2 | L3 | Mean | Context | Pose | Shape | Occlusion | Texture | Weather | Mean |
| | Resnet50 | 33.8 | 22.4 | 15.8 | 9.1 | 15.8 | 15.5 | 12.6 | 15.7 | 16.2 | 22.3 | 23.4 | 18.0 |
| | Swin-T | 29.7 | 23.3 | 15.6 | 10.8 | 16.6 | 18.3 | 14.4 | 16.9 | 17.2 | 21.1 | 26.3 | 19.8 |
| | Convnext | 38.9 | 22.8 | 12.8 | 6.6 | 14.1 | 18.1 | **14.5** | 16.5 | 16.9 | 21.7 | 26.6 | 19.9 |
| $ACC_{\frac{\pi}{18}}$ ↑ | ViT-b-16 | 38.0 | 23.9 | 13.7 | 7.4 | 15.0 | 24.7 | 13.8 | 15.6 | 15.8 | 25.0 | 28.3 | 21.5 |
| | NeMo | 62.9 | **45.0** | **30.7** | **14.6** | **30.1** | 21.9 | 6.9 | 19.5 | **21.7** | **34.0** | 30.4 | **21.9** |
| | Ours | 61.6 | 42.8 | 27.0 | 11.6 | 27.2 | **20.6** | 10.4 | **20.7** | 20.7 | 33.7 | **30.5** | 21.8 |
| | Resnet50 | 82.2 | 66.1 | 53.1 | 42.1 | 53.8 | **57.8** | 34.5 | 50.5 | 53.1 | **61.5** | 60.0 | 51.8 |
| | Swin-T | 81.4 | 58.5 | 47.3 | 38.8 | 48.2 | 52.3 | 41.1 | 45.7 | 48.2 | 50.1 | 64.9 | 50.9 |
| | Convnext | 82.4 | 63.7 | 47.9 | 36.4 | 49.3 | 51.7 | **43.4** | 44.8 | 46.9 | 48.0 | **65.9** | 50.7 |
| $ACC_{\frac{\pi}{6}}$ ↑ | ViT-b-16 | 82.0 | 65.4 | 49.5 | 37.6 | 50.8 | 54.7 | 34.0 | 49.5 | 49.4 | 59.1 | 59.0 | 51.3 |
| | NeMo | 87.4 | 75.9 | **63.9** | **45.6** | **61.8** | 50.3 | 35.3 | 49.6 | **62.9** | 57.5 | 52.2 | 48.0 |
| | Ours | 86.1 | 74.8 | 59.2 | 37.3 | 57.1 | 54.3 | 38.0 | **53.5** | 62.7 | **61.5** | 57.3 | **51.9** |

Table S5: Classification and 3D-pose estimation estimation results for (corrupted)-PASCAL3D+. 3D-pose estimation is evaluated for error under two thresholds: $\frac{\pi}{6}$ and $\frac{\pi}{18}$ separately. Our approach significantly outperforms all baselines for Classification while it reaches SOTA performances for 3D-pose estimation.

| Dataset | | | P3D+ | corrupted-P3D+ | | | | | | | | | | | | |
|---|---|---|---|---|---|---|---|---|---|---|---|---|---|---|---|---|
| Nuisance | | | L0 | defocus blur | glass blur | motion blur | zoom blur | snow | frost | fog | brightness | contrast | elastic transform | pixelate | jpeg compression | Mean |
| Classification $ACC \uparrow$ | | Resnet50 | 99.3 | 67.6 | 41.4 | 73.5 | 87.5 | 84.4 | 84.3 | 93.9 | 98.0 | 90.0 | 46.4 | 82.1 | 95.5 | 78.7 |
| | | Swin-T | **99.4** | 60.7 | 37.1 | 70.9 | 81.3 | 88.5 | 91.6 | 95.4 | 97.9 | 92.1 | 56.3 | 79.2 | 95.3 | 78.9 |
| | | Convnext | **99.4** | 70.1 | 58.7 | 76.5 | **90.0** | **92.3** | 92.9 | **98.5** | **99.2** | **98.4** | 67.6 | 84.2 | **98.7** | 85.6 |
| | | ViT-b-16 | 99.3 | 64.5 | **78.1** | 80.3 | 88.2 | 91.2 | **94.1** | 90.5 | 98.7 | 85.1 | 84.8 | **96.9** | **98.7** | 87.6 |
| | | Ours | 99.1 | **90.1** | 66.9 | 86.8 | 84.9 | 81.3 | 88.1 | 98.2 | 97.9 | 96.8 | **96.7** | **96.9** | 98.1 | 90.2 |
| 3D-pose estimation | $ACC_{\frac{\pi}{6}} \uparrow$ | Resnet50 | 82.2 | 45.6 | 32.2 | 42.3 | 59.6 | 55.4 | 61.4 | 65.3 | 71.3 | 61.5 | 45.6 | 48.8 | 67.1 | 54.7 |
| | | Swin-T | 83.9 | 47.2 | 41.3 | 46.1 | 53.4 | 54.3 | 61.4 | 65.7 | 70.6 | 61.9 | 41.3 | 51.2 | 66.4 | 55.1 |
| | | Convnext | 82.4 | 59.3 | 42.0 | 50.1 | **68.4** | **69.1** | **74.3** | 78.7 | 80.9 | **79.1** | 56.5 | 62.8 | 78.8 | 66.7 |
| | | ViT-b-16 | 81.9 | 45.2 | **54.6** | 53.0 | 63.4 | 60.2 | 69.7 | 63.9 | 78.1 | 60.5 | 56.4 | 72.2 | 78.0 | 62.9 |
| | | NeMo | 87.4 | 71.8 | 54.1 | **68.7** | 65.5 | 60.1 | 69.7 | 81.6 | 81.7 | 77.2 | 76.2 | 73.0 | 78.1 | **71.4** |
| | | Ours | 86.1 | 71.8 | 48.6 | 67.0 | 61.8 | 57.0 | 68.5 | 83.0 | 82.1 | 78.4 | **79.2** | **78.0** | 78.7 | 71.2 |
| | $ACC_{\frac{\pi}{18}} \uparrow$ | Resnet50 | 39.0 | 10.3 | 4.3 | 9.2 | 18.4 | 16.2 | 22.2 | 19.6 | 30.1 | 19.3 | 8.6 | 10.1 | 21.7 | 15.8 |
| | | Swin-T | 46.2 | 13.1 | 7.0 | 8.7 | 12.0 | 12.8 | 17.0 | 21.8 | 27.7 | 21.5 | 6.3 | 14.0 | 25.1 | 15.6 |
| | | Convnext | 38.9 | 16.9 | 12.0 | 12.9 | 23.9 | 24.0 | 29.7 | 32.2 | 37.5 | 35.3 | 17.0 | 17.4 | 30.8 | 24.1 |
| | | ViT-b-16 | 38.0 | 10.1 | 17.5 | 14.2 | 20.7 | 15.8 | 27.2 | 22.6 | 35.5 | 16.5 | 18.8 | 24.3 | 32.5 | 21.3 |
| | | NeMo | 62.9 | 42.9 | **24.6** | **40.2** | **36.7** | **33.0** | **41.2** | 54.6 | 54.5 | 48.9 | 47.9 | 46.6 | 50.2 | 43.4 |
| | | Ours | 61.6 | 44.1 | 22.1 | 38.3 | 33.0 | 29.4 | 40.5 | **56.5** | 55.6 | 51.8 | 52.0 | **50.8** | **51.9** | 43.8 |

### B.4   3D-POSE INITIALIZATION RESULTS

To initiate the render-and-compare process, we require an initial pose denoted as $\alpha_{init}$. We achieve this by pre-sampling 144 distinct feature maps and subsequently calculating the similarity between these extracted features from the image and the pre-rendered maps. The initial pose $\alpha_{init}$ is then determined as the pose corresponding to the highest similarity between the rendered map and the image feature map.

Importantly, by utilizing $\alpha_{init}$ as a coarse 3D pose prediction, we can achieve a remarkable computation speed of approximately 0.04 seconds per sample on a single RTX 3090 GPU. This represents an 80% reduction in computation time compared to the full pipeline of our approach. We oserve in Table S6 that $\frac{\pi}{6}$ results are consistent with the full pipeline. However, the $\frac{\pi}{18}$ results suffer from the coarse prediction importantly. The coarse prediction performs quite well in terms of $\frac{\pi}{6}$ but drops significantly compared to our full pipeline.

Table S6: 3D-Pose Estimation results for different datasets. A prediction is considered correct if the angular error is lower than a given threshold (*i.e.,* $\frac{\pi}{6}$, and $\frac{\pi}{18}$). The coarse prediction performs quite well in terms of $\frac{\pi}{6}$ but drops significantly compared to our full pipeline.

| Dataset | P3D+ | occluded-P3D+ | corrupted-P3D+ | OOD-CV | P3D+ | occluded-P3D+ | corrupted-P3D+ | OOD-CV |
|---|---|---|---|---|---|---|---|---|
| Threshold | | | $\pi/6$ | | | | $\pi/18$ | |
| Ours w/o R&C | 84.3 | 52.8 | 34.2 | 56.7 | 29.8 | 17.0 | 21.1 | 18.8 |
| Ours | 86.2 | 61.9 | 47.9 | 71.5 | 60.5 | 31.6 | 43.4 | 21.8 |

### B.5   ADDITIONAL COMPARISON RESULTS WITH WANG ET AL. (2023)

Our method demonstrates a significant improvement over DMNT (Wang et al., 2023) across various classification scenarios, both in IID and OOD settings. We present detailed classification results in Tables S7 and S8 on PASCAL3D+, occluded-PASCAL3D+, and corrupted-PASCAL3D+. While we couldn't perform a direct comparison on the OOD-CV dataset due to the necessity of training DMNT from scratch, our model excels in handling different types of corruptions, surpassing DMNT by a significant margin—approximately 20 percent overall.

Importantly, our proposed method exhibits efficient real-time classification, averaging a mere 0.02 seconds per image. In contrast, Wang et al. (2023) relies on a brute force search during inference, taking an average of 0.5 seconds per image on the Pascal3D+ dataset under the same setup. This results in a substantial 25 times slower inference, coupled with lower accuracy and robustness.

We employ a vMF-based contrastive loss, fostering distinct features through a dot product operation, while Wang et al. (2023) relies on a Euclidean-distance based loss to estimate feature vector distances. Additionally, our approach distinguishes itself by exploiting individual vertices during inference, as opposed to the collective consideration of vertices as a mesh in both NeMo and DMNT.

It's worth noting that Wang et al. (2023) primarily focuses on obtaining deformable (exact) geometry of objects during inference. In summary, our two approaches differ significantly in both low-level implementation and high-level goals, as well as in terms of computational requirements and classification results. Notably, we refrain from comparing 3D pose estimation performances, given that DMNT exhibits lower performance compared to NeMo in this regard.

Table S7: Classification accuracy results on PASCAL3D+ and occluded-PASCAL3D+ datasets. Higher is better. Our approach outperforms DMNT by a large margin for all nuisances.

| Dataset | P3D+ | occluded-P3D+ | | | |
|---|---|---|---|---|---|
| Nuisance | L0 | L1 | L2 | L3 | Mean |
| DMNT | 94.1 | 85.0 | 67.8 | 43.2 | 65.3 |
| Ours | **99.5** | **97.2** | **88.3** | **59.2** | **81.6** |

Table S8: Classification accuracy results on corrupted-PASCAL3D+ under 12 different types of common corruptions. Our approach outperforms DMNT by a large margin for all corruptions.

| Nuisance | L0 | defocus blur | glass blur | motion blur | zoom blur | snow | frost | fog | bright-ness | con-trast | elastic trans. | pixelate | jpeg | mean |
|---|---|---|---|---|---|---|---|---|---|---|---|---|---|---|
| DMNT | 94.1 | 87.1 | 55.8 | 78.0 | 76.2 | 69.4 | 79.6 | 91.3 | 92.9 | 89.9 | 72.5 | 78.6 | 92.3 | 72.7 |
| Ours | **99.5** | **90.5** | **65.7** | **86.4** | **84.2** | **91.2** | **89.5** | **98.4** | **98.4** | **97.1** | **97.2** | **97.1** | **98.4** | **91.3** |

## B.6 EFFICIENCY

For classification, our method attains real-time performance comparable to other CNN or transformer-based baselines, consistently handling over 50FPS, as detailed in Table S9. Despite variations in parameter numbers among these baselines, we find no discernible correlation between parameter count and OOD robustness. Notably, Table S9 illustrates that our approach outperforms all baselines significantly at OOD generalization, even with similar inference speeds.

In the realm of pose estimation, our model demonstrates a substantial reduction in parameter count when compared to the render-and-compare-based method NeMo (Wang et al., 2021a), as indicated in Table S9. This reduction is attributed to our class-agnostic backbone. Additionally, our inference speed surpasses NeMo by approximately 20 times. This acceleration is primarily due to the fewer steps involved in our render-and-compare process (Ours: 30 steps, NeMo: 300 steps), driven by the observed quicker convergence towards local optima facilitated by our class-contrastive representation. Although CNN and transformer-based baselines exhibit higher inference speeds for pose estimation, their performance in OOD scenarios is notably inferior.

In terms of floating-point operations (FLOPs), Table S9 reveals a stark contrast, with NeMo requiring a significantly higher number of FLOPs at 3619 GFLOPs, whereas our approach demands only 306 GFLOPs. The increased number of operations in our approach can be mainly attributed to the final feature matching procedure in comparison to other CNN and transformer-based methods.

Table S9: Overview of the parameter counts and computation time and cost for each method as well as performances in OOD scenarios (*i.e.,* mean over all nuisances of the OOD-CV dataset). NeMo refers to Wang et al. (2021a). For our approach, we show differnet values for the GFlops, for the classification and pose estimation pipeline, respectively.

| Method | #Parameter | GFlops | Inference speed classification (s) | Inference speed pose estimation (s) | Accuracy classification | Accuracy pose estimation |
|---|---|---|---|---|---|---|
| Resnet50 | 84M | 84.5 | 0.01 | 0.01 | 51.4 | 51.8 |
| Swin-T | 28M | 60.8 | 0.01 | 0.01 | 64.2 | 50.9 |
| Convnext | 30M | 91.1 | 0.01 | 0.01 | 56.0 | 50.7 |
| ViT-b-16 | 86M | 22.6 | 0.01 | 0.01 | 59.0 | 48.0 |
| NeMo | 996M | 3619.2 | NA | 4.02 | NA | 71.4 |
| Ours | 83M | 98.0 - 301.6 | 0.02 | 0.21 | **85.2** | **71.5** |

## B.7 ABLATION STUDY

In the main paper, we conducted ablation experiments exclusively on two datasets, namely Pascal3D+ and occluded-P3D+. This choice was necessitated by computational resource constraints. It is important to note that we do not possess any compelling rationale to believe that the outcomes observed in the aforementioned datasets would significantly differ had we examined the remaining two datasets.

In the main paper, we only evaluated ablations for two datasets (*i.e.,* Pascal3D+ and occluded-P3D+) due to computation resources. We do not have any reason to think that findings made on the aforementioned datasets would be any different if studied on the two remaining datasets.

### B.7.1 SPHERICAL MESH

In order to evaluate our approach with a different mesh, we studied in more details the performances using a spherical mesh. We used the same mesh for all object classes. The mesh is composed of 2562 vertices with 5120 faces. We followed exactly the same approach as the one described in the main paper. We trained our model for 200 epochs using contrastive learning, including the background features. On top of the results provided in the main paper, we provide some additional 3D pose estimation results in Table S10 to compare the performances of both our approach with cuboid and spherical meshes.

Table S10: 3D-Pose Estimation results. A prediction is considered correct if the angular error is lower than a given threshold (*i.e.,* $\frac{\pi}{6}$, and $\frac{\pi}{18}$). Higher is better. We observe that a spherical mesh performs well in IID scenarios but does not generalize as well as the cuboid in more difficult OOD scenarios.

| Threshold | | $\pi/6$ | | | | | $\pi/18$ | | | |
|---|---|---|---|---|---|---|---|---|---|---|
| Dataset | P3D+ | occluded-P3D+ | | | | P3D+ | occluded-P3D+ | | | |
| | L0 | L1 | L2 | L3 | Mean | L0 | L1 | L2 | L3 | Mean |
| Ours w/ sphere | 90.2 | 75.0 | 62.1 | 40.1 | 59.1 | 65.9 | 47.0 | 30.1 | 13.9 | 30.3 |
| Ours (w/ cuboid) | 86.2 | 74.8 | 63.2 | 44.3 | 60.9 | 60.5 | 46.8 | 31.0 | 15.6 | 31.6 |

### B.7.2 "SINGLE VERTEX" ABLATION

To assess the efficacy of our 3D representation, we conducted an ablation study focused on the 3D representation itself. This involved replacing the entire mesh with a single feature vector. In a manner akin to Section 3, we defined a specific mesh $\mathfrak{N}_y$ for this case, where $\mathcal{V}_y$ is represented by $\overrightarrow{0} \in \mathbb{R}^3$, and $\mathcal{C}_y = C \in \mathbb{R}^D$ represents the single vertex feature. Subsequently, we executed a contrastive learning process, randomly selecting positive features from the object in the image and negative features from the image background. Notably, we omitted the background model from consideration since the number of feature vectors aligns with the number of classes (e.g., 12 for P3D+). Consequently, these features are already suitably distributed within the feature space. During the classification inference stage (where predicting the 3D pose is not feasible within the current setup), we computed vertex matching with the 12 vertex features at our disposal. The predicted class corresponds to the class for which the vertex feature exhibits the most significant matching with the image features.

## C ESTIMATION OF CONCENTRATION PARAMETERS

There is no closed form solution to estimating the the concentration ($\kappa$) parameters. Therefore, we set it to $\kappa = 1$. To ensure that setting the values to a constant, we tested the effect of approximating the parameter using a standard method as proposed in (Sra, 2012). In Figure S1, we can observe that the concentration of the learned features is slightly higher, where the object variability is low (e.g. wheels, bottle, ...), whereas it is lower for objects with very variable appearance or shape (e.g. airplanes, chairs, sofa, ...). When integrating the learned concentration parameter into the classification and pose estimation inference we observe almost no effect on the results (see Table S11). Estimating these parameters is non-trivial because of the lack of a closed-form solution and potential imbalances among the visibility of different vertices. This needs to be studied more thoroughly, but our hypothesis is that the learned representation compensates for the mismatch of the cuboid to the object shape and therefore the weighting of the concentration parameter does not have a noticeable effect.

## D ADDITIONAL VISUALIZATIONS

We have generated **supplementary video** visualizations showcasing our Mesh vertex activations, which are available in a separate directory. In the final version of the paper, we will include links to access these videos for a more comprehensive understanding of our approach.

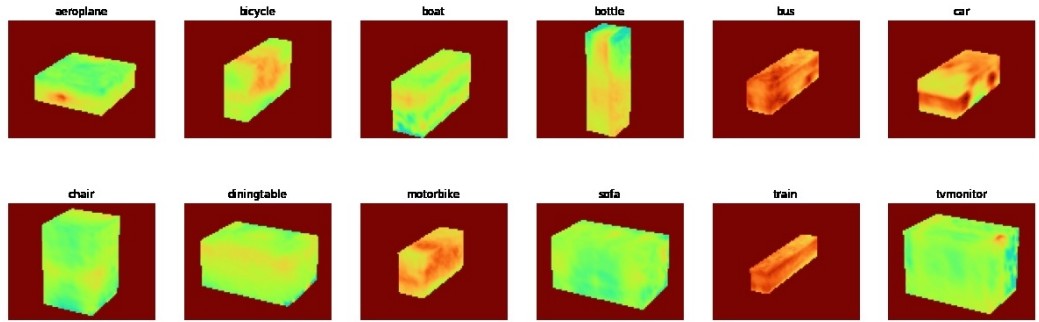

Figure S1: Plot of the concentration $\kappa$ estimates for each vertex using the training dataset (range shown is 0.5 (blue) - 1 (red)).

Table S11: Classification accuracy results on PASCAL3D+ and occluded-PASCAL3D+.

| Dataset | P3D+ | occluded-P3D+ | | | |
|---|---|---|---|---|---|
| Nuisance | L0 | L1 | L2 | L3 | mean |
| Ours with learned $\kappa$ | 99.5 | 97.2 | 88.4 | 59.2 | 81.6 |
| Ours with $\kappa = 1$ | 99.5 | 97.2 | 88.3 | 59.2 | 81.6 |

Additionnaly, in Figure S2, we provide qualitative results. Every image is OOD data with different nuisances. We can see that these scenarios are very likely to be encountered by classification models in the real world. From the correct mesh selected and the correct pose the mesh has aligned, and we can see how our approach successfully predicts both the object category and object pose for these challenging images.

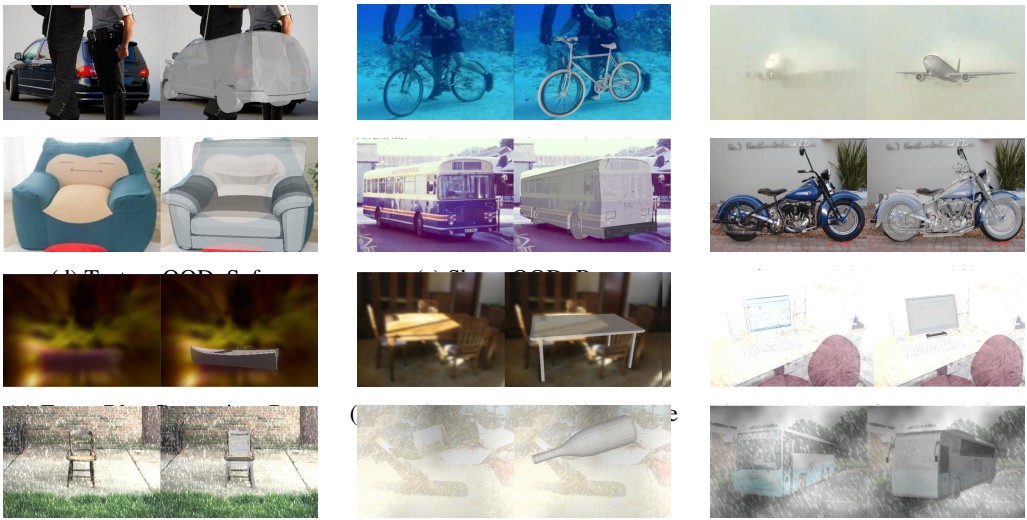

(j) Snow Corruption, Chair    (k) Brightness Corruption, Bottle    (l) Fog Corruption, Bus

Figure S2: Qualitative results of our approach on Occluded PASCAL3D+ and OOD-CV.v2 (a-f), and on Corrupted PASCAL3D+ (g-l). We illustrate the predicted 3D pose using a CAD model. Note that the CAD model is not used in our approach. All images were correctly classified by our approach but incorrectly classified by at least one baseline.

