# OpenReview forum: "3D Object Representation Learning for Robust Classification and Pose estimation"
_ICLR.cc/2024/Conference — Submitted to ICLR 2024_

### Official Review · Reviewer_LehM · 2023-10-26

**Soundness:** 2 fair
**Presentation:** 3 good
**Contribution:** 1 poor
**Rating:** 3
**Confidence:** 3

**Summary:**

The paper proposes a method for joint object classification and 3D pose estimation from a single image.
The core idea is that by doing joint pose estimation and classification, the classification results are more robust in scenarios with challenging occlusion and image corruption.

Each object class's 3D shape is modelled using a dense 3D mesh. Each vertex has a feature descriptor designed to be pose invariant.
The calssification network uses a backbone to detect 2D features, and the next computes the similarity of each 2D feature to the viewpoint-invariant features for each of the 3D vertices of an object class mesh, computing scores (these essentially produces dense 2D-3D correspondences). These scores are used to compute overall per-class scores to perform classification. A set of background features are used to represent the background class. In a second step (not needed for classification), the object pose is computed for the selected object using the 2D-3D dense correspondences using a render-and-compare approach starting from multiple initialisations.

At training time, featrures for 3D vertices and 2D features prameters are compute jointly. Contrastive learning is used to make the 3D features viewpoint invariant (enforce similarity for 3D features corresponding to the same vertex from different viewpoints,  and doing the opposite for 3D features for different vertices or different objects).

The method is evaluated on both object classification on standard benchmarks (Pascal 3D, a heavily occluded version of it, OOD-CV2 for out of distribution samples), and compared against some baselines.

**Strengths:**

- The paper is well written and easy to understand
- The results of the method are convincing, and demonstrate one of the paper claim, i.e. that 3D reasoning makes the approach more robust to occlusions and challenging image conditions.
- The paper is evaluated on standard benchmarks, with reasonable ablation analysis
- The method has some interesting interpretability properties, for example by looking at vertex activation output it is possible to understand which parts of the object are occluded in the image

**Weaknesses:**

My main concern with this paper is lack of novelty, in particular with respect to: "Neural Textured Deformable Meshes for Robust Analysis-by-Syntheses" by Wang et al. (which is cited in the submission). This paper seems to follow the same procedure, using mostly the same techniques (feature extraction, 3D mesh representation, contrastive divergence to make 3D features more invariant, etc.). There seem to be some differences (e.g. it seems the submission the 2D to 3D matches is done differently, and the submission, unlike Wang et al., allows to do classification without the more computationally expensive step of 3D pose estimation, but I am not certain),  but these differences are not explained in the submission, nor seem significant enough to warrant sufficient novelty. Moreover, Wang et al. achieve very comparable results, and in some cases superior performance (e.g. for L3) on Occluded PASCAL3D+ (Compare Table 1 in Wang et al. to Table 1 in the submission). I would have expected a much more detailed comparison to the most relevant papers (in particular "Neural Textured Deformable Meshes for Robust Analysis-by-Syntheses" by Wang et al.), with clear articulation of what the diffrences are, and pros and cons of the proposed method with respect to these alternatives

I also think some claims are not adequately supported by the results, for example "exceptionally robust classification and pose estimation results" as claimed in the abstract

**Questions:**

1) Could you please provide a detailed explanation of the differences with respect to the most similar approaches (all work by Wang et al., in particular Wang et al. 2023)?
2) Is it possible to compare to Wang et al. 2023 on the corrupted Pascal3d+ dataset?

---

> ### Author Response · Authors · 2023-11-17
> **Response to reviewer LehM**
>
> We thank the reviewer for the constructive feedback.
>
> **My main concern with this paper is lack of novelty, in particular with respect to: "Neural Textured Deformable Meshes for Robust Analysis-by-Syntheses" by Wang et al. (which is cited in the submission). This paper seems to follow the same procedure, using mostly the same techniques (feature extraction, 3D mesh representation, contrastive divergence to make 3D features more invariant, etc.). There seem to be some differences (e.g. it seems the submission the 2D to 3D matches is done differently, and the submission, unlike Wang et al., allows to do classification without the more computationally expensive step of 3D pose estimation, but I am not certain), but these differences are not explained in the submission, nor seem significant enough to warrant sufficient novelty. Moreover, Wang et al. achieve very comparable results, and in some cases superior performance (e.g. for L3) on Occluded PASCAL3D+ (Compare Table 1 in Wang et al. to Table 1 in the submission). I would have expected a much more detailed comparison to the most relevant papers (in particular "Neural Textured Deformable Meshes for Robust Analysis-by-Syntheses" by Wang et al.), with clear articulation of what the differences are, and pros and cons of the proposed method with respect to these alternatives**
>
> We are sorry for the confusion with the related paper [G]. We cited the paper for the sake of completeness, and should have better pointed out that it is contemporaneous concurrent work that was **not published and only available on ArXiv** at the time of submission.  We note that according to ICLR reviewer guidelines we should therefore not be required to compare it as a baseline in our experiments (see last answer in the FAQ in [F]). Nevertheless, when comparing our method with [G] the reviewer must have misread the results of the Tables in [G] since our method outperforms [G] significantly at classification and pose estimation in all IID and OOD scenarios. In addition, as the reviewer requested, we contacted the authors and received their code to test on the corrupted-P3D+ dataset. We report the results in Table 4, and can observe that our model outperforms all types of corruption with a large margin, and we are around 20 percent higher overall.
>
> We further want to emphasize that, our proposed method performs classification efficiently in real-time, which takes on average 0.02 seconds per image, whereas [G] simply runs a brute force search during inference which takes on average 0.5 seconds per image on the Pascal3D+ dataset under the same setup, suffering from a 25 times slower inference in addition to the lower accuracy and robustness, and hence making it practically infeasible.
>
> The way how we formulate the contrastive loss is also different, which can be illustrated by the loss function in two papers. We use a vMF-based contrastive loss that encourages features to be distinct through a dot product operation whereas [G] uses a Euclidean-distance based loss estimating the distance between feature vectors.
> The main focus of [G] is different from this paper. Its goal is to obtain deformable (exact) geometry of the object in its forwarding.
> In summary, these two papers have major differences in both low-level implementation and high-level goals, and performance [G] in terms of both accuracy and efficiency for the classification task.
>
> **I also think some claims are not adequately supported by the results, for example "exceptionally robust classification and pose estimation results" as claimed in the abstract**
>
> We will tone down this paragraph in the abstract. However, we note that our proposed model significantly outperforms the best baselines at image classification in OOD scenarios, e.g. +5.8% over ConvNext on occluded-P3D+, +21% over Swin-T on OOD-CV, and +3.7% over ViT-16 on corrupted-Pascal3D+. We also note that the ranking among the baselines varies significantly across datasets, e.g. Swin-T achieves the best ranking among baselines on OOD-CV but only ranks third on corrupted-Pascal3D+. In contrast, our proposed method consistently outperforms the baselines across all OOD scenarios.
>
> **Table 4:** Classification accuracy results on corrupted-PASCAL3D+ under 12 different types of common corruptions. Our approach outperforms DMNT by a large margin for all corruptions.
>
> | Nuisance|L0|defocus blur|glass blur|motion blur|zoom blur|snow|frost|fog|brightness|contrast | elastic trans. | pixelate | jpeg | mean|
> |-|-|-|-|-|-|--|-|-|-|-|-|-|-|-|
> | DMNT|94.1|87.1|55.8|78.0| 76.2|69.4|79.6|91.3|92.9|89.9|72.5|78.6|92.3|72.7|
> |Ours|**99.5**|**90.5**|**65.7**|**86.4**| **84.2**|**91.2**|**89.5**|**98.4**|**98.4**|**97.1**| **97.2**|**97.1**|**98.4**|**91.3**|
>
> [F] Reviewer guide: https://iclr.cc/Conferences/2024/ReviewerGuide
>
> [G] Angtian Wang, Wufei Ma, Alan Yuille, and Adam Kortylewski. Neural textured deformable meshes for robust analysis-by-synthesis. arXiv preprint arXiv:2306.00118, 2023

---

> > ### Comment · Reviewer_LehM · 2023-11-18
> > **Response to rebuttal**
> >
> > The authors's rebuttal allowed me to understand the contributions of the paper better. The authors point out that the reviewer policies do not require comparison to unpublished concurrent work, which is the case for "Neural Textured Deformable Meshes for Robust Analysis-by-Synthesis" - this addresses my main concern on lack of novelty.
> >
> > I have three asks to improve the manuscript:
> > 1. Make it much clearer that not having to do render-and-compare for classification while still exploiting 3D reasoning is a big contribution because it speeds up inference, and that render-and-compare is optional for the pose estimation step, which is not required for classification.This is explained better in the rebuttal than in the new version of the paper.
> > 2. Also the comparison with Nemo is clearer in the rebuttal than in the paper, particularly the points on the class-agnostic backbone, that render-and-compare is not needed for classification, and the increased robustness on OOD. Please make the paper as clear as the rebuttal
> > 3. Similarly, the rebuttal is much more clear on the differences with respect to "Neural Textured Deformable Meshes for Robust Analysis-by-Synthesis". The paper text on this point is too vague.

---

> > > ### Author Response · Authors · 2023-11-20
> > > **Response to reviewer LehM**
> > >
> > > We appreciate your valuable feedback, which prompted us to enhance the manuscript.
> > > We are delighted to have addressed the reviewer’s main concern and clarify the contributions of our paper. Furthermore, we modified our paper according to the feedback of the reviewer. The changes, **highlighted in red**, more accurately emphasize (1) the 3D reasoning in our classification approach without relying on render-and-compare, (2) distinctions from NeMo, and (3) differences from "Neural Textured Deformable Meshes for Robust Analysis-by-Synthesis."
> > >
> > > We appreciate the time and effort the reviewer has put into reviewing our work. We would like to kindly ask the reviewer to reconsider the rating of our paper in light of these improvements. We are fully prepared to address any additional questions or conduct further experiments as suggested. If there are specific aspects of the paper the reviewer believes could be improved, we are ready to respond promptly.

---

### Official Review · Reviewer_35dn · 2023-10-30

**Soundness:** 2 fair
**Presentation:** 2 fair
**Contribution:** 2 fair
**Rating:** 5
**Confidence:** 4

**Summary:**

The paper claims that they propose a 3D object representation at the object category-level that can be used for object classification and 3D object pose estimation.
They represent each object category as a cubic with attached features in each vertex, which are trained using multi-view posed images.
Then, the features are used to classify the object category by directly matching the image feature map with each category's 3D features and estimating the pose of the object in the image using render-and-compare.

**Strengths:**

1) The proposed method uses trained 3D features of each category of object classification, instead of performing 3D object pose estimation only, which is pretty interesting since the 3D features can further be leveraged.

2) Visualizations are attached to show the interpretability of the 3D features.

**Weaknesses:**

1) More detailed discussions and comparisons with NeMo. NeMo is highly related to the proposed paper, but the discussion is missing in the introduction and related works. As far as I can see, the 3D object representation in the paper is already proposed by NeMo. NeMo already finds that using a cubic for an object class can achieve good performance in 3D pose estimation. They also use contrastive loss for training and render-and-compare strategy for pose estimation as in this paper. I think the difference is that the paper extends this existing 3D representation to training on multiple categories and uses it for classification. Based on this point, I think a huge part of the contributions of this paper belong to NeMo, and the real contributions are limited.

2) For the evaluation of classification, I think the paper should compare with the state-of-the-art 2D object detectors for classification to show the advantages. For example, I think the YOLO can be easily trained using the same training data as the proposed method, i.e., the projected 2D bounding boxes from the 3D cubics.

**Questions:**

Please refer to the weaknesses.

---

> ### Author Response · Authors · 2023-11-17
> **Response to reviewer 35dn**
>
> We thank the reviewer for the constructive feedback.
>
> **More detailed discussions and comparisons with NeMo. NeMo is highly related to the proposed paper, but the discussion is missing in the introduction and related works. As far as I can see, the 3D object representation in the paper is already proposed by NeMo. NeMo already finds that using a cubic for an object class can achieve good performance in 3D pose estimation. They also use contrastive loss for training and render-and-compare strategy for pose estimation as in this paper. I think the difference is that the paper extends this existing 3D representation to training on multiple categories and uses it for classification. Based on this point, I think a huge part of the contributions of this paper belong to NeMo, and the real contributions are limited.**
>
> Our method indeed builds on several previous works while substantially extending them.
> Compared to NeMo, we make several key contributions: (1) An architecture with a class-agnostic backbone, while NeMo uses a separate backbone for each object. This enables us to introduce a class-contrastive loss that enables object classification, while NeMo is not capable of classifying images. As illustrated in the t-SNE plots in reviewer-wnbh’s answer. (2) A principled way of speeding up the inference without having to perform render-and-compare, which advances the brute-force testing and optimization of other concurrent methods [G] (see answer to reviewer-lehm for more details). (3) A comprehensive mathematical formulation that derives a vMF-based contrastive training loss that is different from Euclidean-distance based contrastive loss of NeMo. (4) We demonstrate the possibility to exploit individual vertices during inference rather than considering vertices collectively as a mesh in NeMo, opening potentials tackling segmentation or object part detection tasks. Finally, all our advances lead to (5) substantial improvements in terms of OOD robustness over all baselines at image classification, while at the same time performing on par with models that were specifically designed for robust pose estimation, such as NeMo.
>
> **For the evaluation of classification, I think the paper should compare with the state-of-the-art 2D object detectors for classification to show the advantages. For example, I think the YOLO can be easily trained using the same training data as the proposed method, i.e., the projected 2D bounding boxes from the 3D cubics.**
>
> Thank you for the suggestion. We trained YOLO using the official repository [D] on the Pascal3D+ data with standard data augmentation in terms of scale, rotation and mirroring data augmentation. From the table below, we can observe that YOLO obtains similar performances for IID data, but suffers even more than the other classification baselines under out-of-distribution shifts in terms of occlusion, whereas our proposed model achieves much stronger robustness. While refining our YOLO results through additional investigation might be possible, we speculate that utilizing the official training code for YOLO might lead to potential overfitting, given its original focus on detection rather than classification. We anticipate that more advanced data augmentation (such as CutOut or CutPaste) techniques could enhance generalization in YOLO's performance. Notably, it is crucial to emphasize that our method, in contrast, does not necessitate data augmentation and demonstrates effective generalization.
>
> **Table 3:** Classification results on P3D+ and occluded-P3D+ comparing Yolo and our approach.
>
> | Nuisance  | L0   | L1                | L2                | L3                | Mean              |
> |-----------|------|-------------------|-------------------|-------------------|-------------------|
> | YOLO      | 99.3 | 62.4              | 44.9              | 25.0              | 44.1              |
> | Ours      | **99.5** | **97.2**        | **88.3**          | **59.2**          | **81.6**          |
>
> [D]: https://docs.ultralytics.com/

---

> > ### Comment · Reviewer_35dn · 2023-11-22
> >
> > Thanks for the clarifications. I acknowledge that the paper has multiple improvements over NeMo that make it class-agnostic and extra improvements. But I think the paper is a little overclaimed that "it proposes a new representation ...", which has already been proposed by NeMo and adapted by this paper.

---

> > > ### Author Response · Authors · 2023-11-22
> > > **Response to reviewer 35dn**
> > >
> > > We appreciate the reviewer's helpful feedback, as well as the time and effort the reviewer has put into reviewing our work. We are happy that the concerns of the reviewer have been addressed. We also toned down the claim about introducing a new representation in the paper (**see changes in purple** in the newest revision of the manuscript).
> > >
> > >
> > > While the wording of our contribution in the abstract was not optimal, we want to further emphasize that our **new class-agnostic backbone** and the **novel class-contrastive loss** lead to a representation that goes beyond the original representation in NeMo, as it leads to a much better disentanglement of the class information, while at the same time disentangling the 3D pose (see T-SNE plots provided to reviewer wnbh). Additionally, our representation used for classification is significantly different from NeMo, as we do not rely on the 3D mesh during classification and hence do not require the expensive render-and-compare optimization for classification.

---

### Official Review · Reviewer_Yc1n · 2023-10-31

**Soundness:** 3 good
**Presentation:** 3 good
**Contribution:** 3 good
**Rating:** 6
**Confidence:** 4

**Summary:**

This paper proposes a novel 3D object representation learning method for robust classification and pose estimation, exploring the establishment of dense correspondences between image pixels and 3D template geometry. The feature of a pixel in a 2D image is mapped to the corresponding vertex in a set of pre-defined 3D template meshes, which are further trained via contrastive learning and associated camera poses for classification. Finally, the poses are estimated by the refinement from the initial pose of the template.

**Strengths:**

1.	The motivation of this work, which is to learn representation from 3D template geometry, is technically sound and fits well into object representation learning.
2.	The design of the inference pipeline is highly efficient, where image classification can be achieved merely using vertex features.
3.	Extensive experiments verify the effectiveness of the proposed approach on 3D object representation learning and classification, and the accurate pose estimations further demonstrate the interpretability.

**Weaknesses:**

1.	All typos should be checked and corrected to improve writing quality (e.g., in the first paragraph of the Introduction Section, "… gradient-based optimization on a specific training set (Figure ??).").
2.	There is a lack of an efficiency comparison with existing methods. It seems that the high efficiency in classification is a critical contribution of this work. A comparison of the inference speed and the number of parameters between the proposed framework and other methods can further support this claim.

**Questions:**

Please refer to the weaknesses listed above.

---

> ### Author Response · Authors · 2023-11-17
> **Response to reviewer Yc1n**
>
> We thank the reviewer for the constructive feedback.
>
> **All typos should be checked and corrected to improve writing quality (e.g., in the first paragraph of the Introduction Section, "… gradient-based optimization on a specific training set (Figure ??).").**
>
> We are sorry for the typos and will revise the manuscript thoroughly.
>
> **There is a lack of an efficiency comparison with existing methods. It seems that the high efficiency in classification is a critical contribution of this work. A comparison of the inference speed and the number of parameters between the proposed framework and other methods can further support this claim.**
>
> In terms of classification, our method achieves similar real-time performance as the other baselines, processing over 50 images per second. Among the baselines there is a large variation in terms of the number of parameters (Resnet: 84M, Swin-T: 28M, ViT-b-16: 86M), but we do not observe any correlation with OOD robustness. Compared to those, our model contains 83M which is comparable. However, concerning the pose estimation baseline Nemo, we have a much lower parameter count (83M vs 996M) and faster inference speed. We will make sure to emphasize this more in the revised paper.

---

> > ### Comment · Reviewer_Yc1n · 2023-11-21
> >
> > I thank the authors for their response. They address most of the points about writing quality and efficiency raised in my review, and I will keep my positive rating.
> >
> > To improve the manuscript, it would be nice if the authors could explicitly compare the number of parameters, FLOPS, and inference speed with baseline methods in the revised version to stress their efficiency advantage.

---

> > > ### Author Response · Authors · 2023-11-22
> > > **Response to reviewer Yc1n**
> > >
> > > We appreciate the reviewer's helpful feedback, as well as the time and effort the reviewer has put into reviewing our work. In response to the latest comments, we've made enhancements to our manuscript, marking the **changes in orange** (starting on page 8). Notably, we put more emphasis on the efficiency of our method, both in terms of memory and computation requirements. Moreover, we added an extensive comparison in the supplementary material.
> > >
> > >
> > > We are ready to address any remaining concerns the reviewer may have. We are committed to enhancing the paper through our dedicated efforts and would appreciate it if the reviewer considers raising the paper's rating further. We once again express our gratitude to the reviewer for the valuable input.

---

### Official Review · Reviewer_wnbh · 2023-10-31

**Soundness:** 3 good
**Presentation:** 3 good
**Contribution:** 2 fair
**Rating:** 6
**Confidence:** 3

**Summary:**

Authors embed 3D geometry into learning a model for image classification.
Important part is identification between background and class-object features, this is done by contrastive repr. learning.

The method uses the fact that objects in the image are constructed from real 3D. The 3D is represented as a mesh of faces and vertices.  The authors learn matching between image features and their corresponding vertex location.

During the interference, the mesh is “rendered and compared” to minimize the likelihood of features.

Various classification experiments are performed. In addition, the 3d pose estimate is evaluated too.

**Strengths:**

Combining 2D and 3D for recognition is a great way to tackle classification. It opens many options for object representation.

The method looks clear. Although I did not check all the equations, the paper is written in way so it is understandable.

Many experiments (maybe too much) are presented. I especially like Sec 4.4.

**Weaknesses:**

Authors start the paper with the statement: “we pioneer a framework for 3D object representation..”, the statement is in contrast with various papers that investigate object representation using techniques on how to map image features to 3D models of faces and vertices. For example M1 or M2.

Dataset creation - details in questions.

Baselines and 3D data - details in questions.

For example, missing papers:
M1: Choy et al. Enriching Object Detection with 2D-3D Registration and Continuous Viewpoint Estimation CVPR 15
M2: Start et al. Back to the Future: Learning Shape Models from 3D CAD Data
 BMVC 10

**Questions:**

\kappa (eq. 3) is fixed to constant, how is the parameter fixed? Is it from validation data, guess, tuning on testing data?

Does baselines use 3D? (It looks that authors use 3D but baselines don't).

Im puzzled in the dataset creation. Originally, pascal authors claim that 10812 testing images from Pascal3D+ are used to create Ocluded PASCAL3D (from official github page). Authors use 10812 Pascal3D+ images as a validation set. Is the authors validation imageset same as the testset in Ocluded Pascal? If so, then it does look like a bug in dataset creation. This question and the question above are look for me as crucial for deciding between acceptance or rejection.

However the conference is called Conf. on Learning Representation. The paper looks to be structured for a computer vision conference. For example, I would be more interested in how the proposed representation (and its modification) affects the result of how the method expresses the objects rather than many classification results that are provided (I'm not saying skip them all) and are better suited for cvpr-like venues. I would like to see some examples where the representation is beneficial to other methods and where it is in contrast worse.

---

> ### Author Response · Authors · 2023-11-17
> **Response to reviewer wnbh (1/2)**
>
> We thank the reviewer for the constructive feedback.
>
> **Authors start the paper with the statement: “we pioneer a framework for 3D object representation..”, the statement is in contrast with various papers that investigate object representation using techniques on how to map image features to 3D models of faces and vertices. For example M1 or M2.**
>
> We acknowledge the relevance of these papers and their techniques in mapping image features to 3D models. We will revise our claim in a more nuanced manner to better align with the state of the art. Additionally, we will incorporate references to M1 and M2 in our related work section, recognizing their significance in the exploration of object representation techniques.
>
> **$\kappa$ (eq. 3) is fixed to constant, how is the parameter fixed? Is it from validation data, guess, tuning on testing data?**
>
> Originally, we did not explore estimating the concentration parameter $\kappa$ because there is no closed form solution to it. Therefore, we set it to $\kappa=1$. Estimating these parameters is non-trivial because of the lack of a closed-form solution and potential imbalances among the visibilities of different vertices. Nevertheless, motivated by the comment of Reviewer-wnbh, we now tested the effect of approximating the parameter using the method as proposed in [A]. In the [Figure 1](https://drive.google.com/file/d/1-6IBsM2FGd84p2r54OPj4cSJoDQUBnwv/view?usp=share_link), we can observe that the concentration of the learned features is slightly higher, where the object variability is low (e.g. wheels, bottle body, …), whereas it is lower for objects with very variable appearance or shape (e.g. airplanes, chairs, sofa, …). When integrating the learned concentration parameter into the classification and pose estimation inference we observe almost no effect on the results (see Table 1). This needs to be studied more thoroughly, but our hypothesis is that the learned representation compensates for the mismatch of the cuboid to the object shape and therefore the weighting of the concentration parameter does not have a noticeable effect.
>
> **Table 1**: Classification results on P3D+ and OCC- P3D+ comparing our results by including and excluding the concentration parameters from the classification during inference.
> | Nuisance  | L0   | L1  | L2| L3  | Mean  |
> |-------------------------------|------|-------------------|-------------------|-------------------|-------------------|
> | Ours with learned $\kappa$    | 99.5 | 97.2    | 88.4              | 59.2              | 81.6              |
> | Ours with $\kappa=1$| 99.5 | 97.2              | 88.3              | 59.2              | 81.6              |
>
>
> **Do the baselines use 3D? (It looks that authors use 3D but baselines don't).**
>
> To the best of our knowledge, our method is the first in the literature to use a 3D representation for image classification. Moreover, it is not clear how to provide existing architectures with additional 3D annotation. Based on the feedback of reviewer-wnbh, we conducted an experiment where we extended several of the classification baselines with an additional pose estimation head. In particular, we follow the popular approach of casting the pose estimation problem as bin classification by discretizing the pose space [C]. We train the models with both the image classification and pose estimation loss until convergence. As can be observed from the table below, the additional 3D information does not improve the classification performance. Moreover, this claim is supported by the t-SNE plots shown in [Figure 2](https://drive.google.com/file/d/1QYy-ExWpWtytRur3oK53yxTSEg951wud/view?usp=share_link). Simply adding 3D-pose information via an additional branch does not induce the network to learn any 3D-aware structure in the representation, whereas the 3D-aware structure is directly built-in in our approach, thus leading to large benefits in terms of robustness and interpretability.
>
> If the reviewer has any alternative baselines to suggest that leverage the advantages of 3D, we are open to incorporating and testing them in our evaluation.
>
>
> **Table 2:** Classification results on P3D+ and occluded-P3D+ comparing resnet baseline performances by training a network per task and a single network for both. We don’t observe any synergy in the unique model trained on both tasks.
>
> | Nuisance | L0    | L1  | L2  | L3 | Mean  |
> |------------------|------|----------|---------|-----|----|
> | Resnet (cls-only)| **99.3** | 93.8  | **77.8**| **45.2** | **72.3** |
> | Resnet (cls&pose)| 99.2 | **93.9** | 77.6  | 45.0 | 72.2  |
>
> [A] Sra, S. (2011). "A short note on parameter approximation for von Mises-Fisher distributions: And a fast implementation of I_s(x)". Computational Statistics. 27: 177–190
>
> [B] Xingyi Zhou, Arjun Karpur, Linjie Luo, and Qixing Huang. Starmap for category-agnostic keypoint and viewpoint estimation. In Proceedings of the European Conference on Computer Vision (ECCV), pp. 318–334, 2018.

---

> > ### Author Response · Authors · 2023-11-17
> > **Response to reviewer wnbh (2/2)**
> >
> > **Im puzzled in the dataset creation. Originally, pascal authors claim that 10812 testing images from Pascal3D+ are used to create Occluded PASCAL3D (from official github page). Authors use 10812 Pascal3D+ images as a validation set. Is the authors validation imageset same as the testset in Occluded Pascal? If so, then it does look like a bug in dataset creation. This question and the question above are look for me as crucial for deciding between acceptance or rejection.**
> >
> > We are sorry for the confusion, this is a mistake in the manuscript. The word “validation” should be replaced with “testing”. We download and use the dataset as released by [C] where the test data from Pascal3D+ was corrupted with random occluders to generate the occluded-Pascal3D dataset.
> >
> > **I would like to see some examples where the representation is beneficial to other methods and where it is in contrast worse.**
> >
> > Based on the request, we present the T-SNE in [Figure 2](https://drive.google.com/file/d/1QYy-ExWpWtytRur3oK53yxTSEg951wud/view?usp=share_link) and [Figure 3](https://drive.google.com/file/d/10oM2QT8RpnxfmyIsOLBU4EL9PfzPs5H-/view?usp=share_link), illustrating the feature representations of ResNet, NeMo, and our approach.
> >
> > In the context of classification, by looking at [Figure 2](https://drive.google.com/file/d/1QYy-ExWpWtytRur3oK53yxTSEg951wud/view?usp=share_link), it becomes evident that our method, much like any classification baseline, results in well-defined clusters for the classes. Conversely, with NeMo, there is significantly less clear clustering.
> >
> > Regarding pose information in the representation, we observe in [Figure 3](https://drive.google.com/file/d/10oM2QT8RpnxfmyIsOLBU4EL9PfzPs5H-/view?usp=share_link) (only car samples from the test dataset are shown) that we can easily extract information about the 3D pose of the object (we only show the azimuth angle for illustration purposes) while a ResNet model that has been trained on both classification and pose estimation task does not have such information explicitly encoded.
> >
> > Therefore, the plots confirm that our approach exhibits class-aware features, enabling high classification performances while simultaneously incorporating the 3D pose awareness into the representation.
> >
> > [C] Angtian Wang, Yihong Sun, Adam Kortylewski, and Alan L Yuille. Robust object detection under occlusion with context-aware compositionalnets. In Proceedings of the IEEE/CVF Conference on Computer Vision and Pattern Recognition, pp. 12645–12654, 2020.

---

> > > ### Comment · Reviewer_wnbh · 2023-11-20
> > >
> > > I would like to thank authors for the clarification. Thanks for all sections that are in the rebuttal, I read them all.
> > >
> > > My main concern was dataset creation mistake that has been clarified. Thus, I stay with my rating: "above the acceptance threshold".
> > >
> > > I would like to comment on section: "I would like to see some examples where the representation is beneficial to other methods and where it is in contrast worse.":
> > >
> > >    - Are there any cases where your method does not work and has weak points?
> > >
> > >   - Figure3: I'm not sure if I understand it correctly. It is written that the left figure ResNet "has been trained on both classification and pose estimation". However, the obtained results suggest that the model did not learn any pose (azimuth).
> > > Fig.3: It is related to Tab.2: Resnet (cls-only) vs. Resnet (cls&pose).  Is it true that this can be seen as a "naive" test to add pose into classification and it does not work in this way?
> > >
> > >   - Fig.2 is nice comparison with clear result.
> > >
> > >   - Kappa explanation is clear.
> > >
> > > Once again, thank you for the rebuttal, the rest is clear. Notes here are details. As I said, I stay with rating #6: Above the acceptance threshold.

---

> > > > ### Author Response · Authors · 2023-11-20
> > > > **Response to reviewer wnbh**
> > > >
> > > > We are very happy to read the positive feedback of the reviewer and that the concerns of the reviewer were resolved. Regarding the follow-up questions:
> > > >
> > > > **Are there any cases where your method does not work and has weak points?**
> > > >
> > > > For the datasets that we tested, we do not observe any specific weak points with respect to the baselines, and rather observe a superior performance across all aspects (accuracy, explainability, robustness). However, we believe that our approach, because of the rigid cuboid representation, may face challenges with strongly articulated or deformable objects, such as humans or animals. Additionally, the coarse cuboid representation poses limitations for tasks that demand high local precision, such as part detection or segmentation, making a direct application to these tasks unrealistic. Addressing these challenges is a very interesting future research direction.
> > > >
> > > >
> > > > **It is written that the left figure ResNet "has been trained on both classification and pose estimation". However, the obtained results suggest that the model did not learn any pose (azimuth). … . Is it true that this can be seen as a "naive" test to add pose into classification and it does not work in this way?**
> > > > Indeed, our experiment of treating 3D pose estimation as a classification problem is a straightforward “naive” extension of existing architectures (and it serves as a standard baseline in the pose estimation literature, e.g. [B]). What is interesting about the result we report is that the ResNet baseline demonstrates a good ability to predict pose (>80%  in IID pi/6 error shown in Table 2 of the paper), but the model does not learn to disentangle the pose information in the representation. This indicates that the model probably focuses on spurious correlations in the data, which ultimately makes it much less robust compared to the disentangled representation in our model.
> > > >
> > > > [B] Xingyi Zhou, Arjun Karpur, Linjie Luo, and Qixing Huang. Starmap for category-agnostic keypoint and viewpoint estimation. In Proceedings of the European Conference on Computer Vision (ECCV), pp. 318–334, 2018.
> > > >
> > > > We appreciate the time and effort the reviewer has put into reviewing our work. We are fully prepared to address any additional questions or conduct further experiments as suggested. If there are specific aspects of the paper the reviewer believes could be improved, we are ready to respond promptly. We hope that with our dedicated efforts, we can enhance the paper and would be happy if the reviewer considers increasing the rating of the paper further.

---

### Author Response · Authors · 2023-11-17
**Response to all reviewers**

We thank all reviewers for the thorough review of our manuscript and the valuable feedback provided. As several reviewers raised concerns regarding the relation of our work to existing pose estimation models, we would like to address these concerns and clarify the focus and contributions of our work.

Our primary emphasis in this work is object classification, and the core contribution of our work lies in introducing a classification method that has 3D knowledge about objects and therefore is highly robust and interpretable, while also being capable of real-time inference, hence addressing a critical need in the current state-of-the-art landscape. We want to emphasize that the ability to perform pose estimation is a byproduct of the origin of the representation and is not part of our contributions, even though the pose estimation results obtained with our method are on par with similar methods designed for pose estimation only.

We address the individual concerns regarding related work on pose estimation, as well as other concerns in the individual rebuttals to each reviewer. We are more than happy to further feedback, questions and to conduct additional experiments for the reviewers.

We also submit a revised manuscript that already reflects changes based on the reviewer feedback (changes are highlighted in blue).
**As a summary, we have made the following changes in our manuscript:**
- We fix the typos in the paper (*Reviewer Yc1n*)
- **Abstract**: We toned down our claims of pioneering 3D representations (*Reviewer wnbh*), emphasized the relationship with pose estimation method NeMo and made clearer what our contributions are.
- **Related work**: We added M1 and M2 papers (*Reviewer wnbh*). We mentioned in more detail the difference with NeMo (*Reviewer 35dn*) . We mention in more detail Wang et al., 2023 and how it differs (*Reviewer LehM*).
- **Experiments**: We corrected the mention to validation data (*Reviewer wnbh*).  We mention kappa experiments that have been added in the Appendix (*Reviewer wnbh*). We merged results from Table 1-2 into Table 1 only (full results are now in Appendix). We comprehensively modified section 4.4 (e.g. tSNE plots, efficiency) (*Reviewer wnbh and Yc1n*)
- **Supplementary**: We added more content about $\kappa$ estimation and additional classification results. (*Reviewer wnbh*)

We personally and directly responded to the feedback of each reviewer below, all Figures we refer to can be found either in the paper or in an anonymous [Google Drive](https://drive.google.com/drive/folders/1DUNjV6tg08ADvzhV9wVHMurtoOXE5836?usp=sharing).

---

> ### Author Response · Authors · 2023-11-22
> **Response to all reviewers**
>
> We are pleased that our efforts to provide clarification have successfully addressed all concerns of all reviewers that were raised throughout the discussion period. We have meticulously integrated all feedback from the reviewers already in the revised manuscript.
>
>
> We express our sincere gratitude for the thoughtful consideration and constructive feedback provided during the discussion process.

---

### Meta-Review · Area_Chair_BqtS · 2023-12-05

**Metareview:**

This paper describes a new 3D object representation learning method for robust classification and pose estimation. The 3D object representation is created at the object category-level that can therefore be used for object classification and 3D object pose estimation.

The strengths of this paper are:
(1) The proposed method uses trained 3D features of each category of object classification instead of performing 3D object pose estimation only so that the 3D features can further be leveraged.
(2) The proposed method has some interesting interpretability properties, for example, by looking at vertex activation output, it is possible to understand which parts of the object are occluded.

On the other hand, the chief weakness is that its deviation from NeMo is rather marginal, although the proposed method makes improvements in a few aspects.

Overall, while the reviewers appreciate the work and quality results, there remains a concern about the novelty of the work. For the reviewer LehM's rating, ACs have taken it into account as it was upgraded. The reviewers and AC read the rebuttal and took it into consideration for the final recommendation.

**Justification For Why Not Higher Score:**

Although the reviewers and AC see the strength of the paper, its novelty is somewhat lacking due to the previous related works, particularly NeMo. There was a discussion between the authors and reviewers, but the concern was not entirely addressed during the rebuttal.

**Justification For Why Not Lower Score:**

N/A

---

### Decision · Program_Chairs · 2024-01-16

Reject